# Flow-GRPO:
# Training Flow Matching Models via Online RL

**Jie Liu**[1,3,5*]   **Gongye Liu**[2,3*]   **Jiajun Liang**[3]   **Yangguang Li**[1]
**Jiaheng Liu**[4]   **Xintao Wang**[3]   **Pengfei Wan**[3]   **Di Zhang**[3]   **Wanli Ouyang**[1,5†]

[1]MMLab, CUHK        [2]Tsinghua University        [3]Kling Team, Kuaishou Technology
[4]Nanjing University        [5]Shanghai AI Laboratory
`jieliu@link.cuhk.edu.hk`   `wlouyang@ie.cuhk.edu.hk`
Code: https://github.com/yifan123/flow_grpo

## Abstract

We propose Flow-GRPO, the first method to integrate online policy gradient reinforcement learning (RL) into flow matching models. Our approach uses two key strategies: (1) an ODE-to-SDE conversion that transforms a deterministic Ordinary Differential Equation (ODE) into an equivalent Stochastic Differential Equation (SDE) that matches the original model's marginal distribution at all timesteps, enabling statistical sampling for RL exploration; and (2) a Denoising Reduction strategy that reduces training denoising steps while retaining the original number of inference steps, significantly improving sampling efficiency without sacrificing performance. Empirically, Flow-GRPO is effective across multiple text-to-image tasks. For compositional generation, RL-tuned SD3.5-M generates nearly perfect object counts, spatial relations, and fine-grained attributes, increasing GenEval accuracy from 63% to 95%. In visual text rendering, accuracy improves from 59% to 92%, greatly enhancing text generation. Flow-GRPO also achieves substantial gains in human preference alignment. Notably, very little reward hacking occurred, meaning rewards did not increase at the cost of appreciable image quality or diversity degradation.

## 1   Introduction

Flow matching [2, 3] models have become dominant in image generation [4, 5] due to their solid theoretical foundations and strong performance in producing high quality images. However, they often struggle with composing complex scenes involving multiple objects, attributes, and relationships [6, 7], as well as text rendering [8]. At the same time, online reinforcement learning (RL) [9] has proven highly effective in enhancing the reasoning capabilities of large language models (LLMs) [10, 11]. While previous research has mainly focused on applying RL to early diffusion-based generative models [12] and offline RL techniques like direct preference optimization [13] for flow-based generative models [14, 15], the potential of online RL in advancing flow matching generative models remains largely unexplored. In this study, we explore how online RL can be leveraged to effectively improve flow matching models.

Training flow models with RL presents several critical challenges: (1) Flow models rely on a deterministic generative process based on ODEs [3], meaning they cannot sample stochastically during inference. In contrast, RL relies on stochastic sampling to explore the environment, learning by trying different actions and improving based on rewards. *This need for stochasticity in RL conflicts with the deterministic nature of flow matching models.* (2) Online RL depends on efficient sampling

---

[*]Equal contribution, [†]Corresponding author

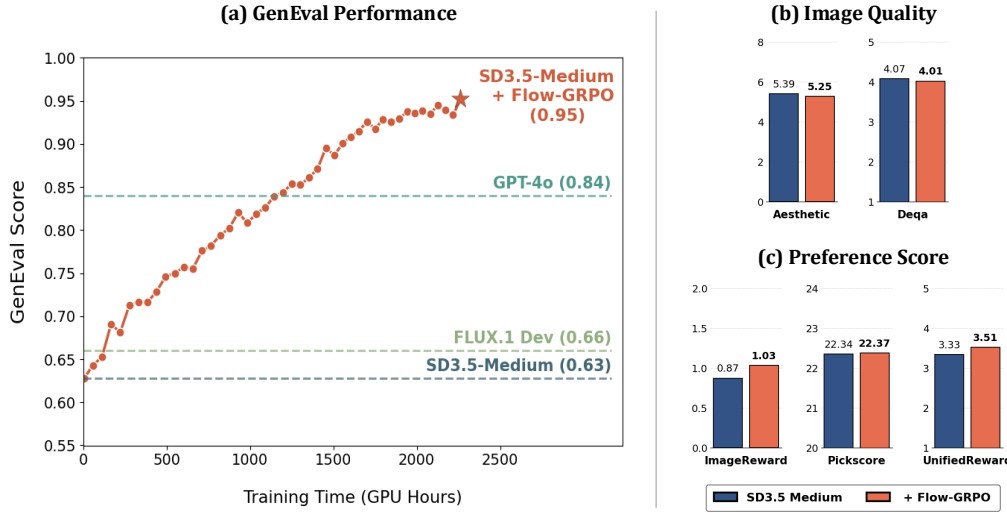

Figure 1: **(a) GenEval performance** rises steadily throughout Flow-GRPO's training and outperforms GPT-4o. **(b) Image quality metrics** on DrawBench [1] remain essentially unchanged. **(c) Human Preference Scores** on DrawBench improves after training. Results show that *Flow-GRPO enhances the desired capability while preserving image quality and exhibiting minimal reward-hacking.*

to collect training data, but flow models typically require many iterative steps to generate each sample, limiting efficiency. This issue is more pronounced with large models [5, 4]. To make RL practical for tasks like image or video generation, *improving sampling efficiency is essential*.

To address these challenges, we propose **Flow-GRPO**, which integrates GRPO [16] into flow matching models for text-to-image (T2I) generation, using two key strategies. First, we adopt the **ODE-to-SDE** strategy to overcome the deterministic nature of the original flow model. By converting the ODE-based flow into an equivalent Stochastic Differential Equation (SDE) framework, we introduce randomness while preserving the original marginal distributions. Second, to improve sampling efficiency in online RL, we apply the **Denoising Reduction** strategy, which reduces denoising steps during training while keeping the full schedule during inference. Our experiments show that using fewer steps maintains performance while significantly reducing data generation costs.

We evaluate Flow-GRPO on T2I tasks with various reward types. (1) Verifiable rewards, using the GenEval [17] benchmark and visual text rendering task. GenEval includes compositional image generation tasks (e.g., generating specific object counts, colors, and spatial relationships), which can be automatically assessed with object detection methods. Flow-GRPO improves the accuracy of Stable Diffusion 3.5 Medium (SD3.5-M) [4] from 63% to 95% on GenEval, outperforming the state-of-the-art GPT-4o [18] model. For visual text rendering, SD3.5-M's accuracy increases from 59% to 92%, greatly enhancing its text generation ability. (2) Model-based rewards, such as the human preference Pickscore [19] reward. These results show that our framework is task independent, demonstrating its generalizability and robustness. Importantly, all improvements are achieved with very little reward hacking, as demonstrated in Figure 1.

To summarize, the contributions of Flow-GRPO are as follows:

- We are the first to introduce GRPO to flow matching models by converting deterministic ODE sampling into SDE sampling, showing the effectiveness of online RL for T2I tasks. Flow-GRPO improves SD3.5-M accuracy from 63% to 95% without noticeably compromising image quality.

- We find that online RL for flow matching models does not require the standard long timesteps for training sample collection. By using fewer denoising steps during training and retaining the original steps during testing, we can significantly accelerate the training process.

- We show that the Kullback-Leibler (KL) constraint effectively prevents reward hacking, where reward increases at the cost of image quality or diversity. KL regularization is not empirically equivalent to early stopping. With a proper KL term, we can match the high reward of the KL-free version while preserving image quality, albeit with longer training.

## 2 Related Work

**RL for LLM.** Online RL has effectively improved the reasoning abilities of LLMs, such as DeepSeek-R1 [10] and OpenAI-o1 [11], using policy gradient methods like PPO [20] or value-free GRPO [16]. GRPO is more memory efficient by removing the need for a value network, so we adopt it in this work. PPO can also be applied to flow matching in a similar way.

**Diffusion and Flow Matching.** Diffusion models [21, 22, 23] add Gaussian noise to data and train a neural network to reverse the process. Sampling uses discrete DDPM steps or probability flow SDE solvers to generate high-fidelity outputs. Flow matching [2, 3] learns a continuous-time normalizing flow by directly matching the velocity field, allowing efficient deterministic sampling with only a few ODE steps. It achieves competitive FID with far fewer denoising steps than diffusion, making it the dominant choice in recent image [4, 5] and video [24, 25, 26, 27] generation models. Recent work [28, 29] unifies diffusion and flow models under an SDE/ODE framework. Our work builds on their theoretical foundations and introduces GRPO to flow-based models.

**Alignment for T2I.** Recent efforts to align pretrained T2I models with human preferences follow five main directions: (1) direct fine-tuning with differentiable rewards [30, 31, 32, 33]; (2) Reward Weighted Regression (RWR) [34, 35, 36, 37]; (3) Direct Preference Optimization (DPO) and variants [38, 39, 14, 40, 41, 42, 43, 44, 45, 46]; (4) PPO-style policy gradients [47, 48, 49, 50, 51, 52]; (5) training-free alignment methods [53, 54, 55]. These methods have successfully aligned T2I models with human preferences, improving aesthetics and semantic consistency. Building on this progress, we introduce GRPO for flow matching models, the backbone of today's state-of-the-art T2I systems. Concurrent work [56] applies GRPO to text-to-speech flow models, but instead of converting the ODE to an SDE to inject stochasticity, they reformulate velocity prediction by estimating a Gaussian distribution (predicting both the mean and variance of velocity), which requires retraining the pre-trained model. Another study [57] also explores SDE-based stochasticity but focuses on inference-time scaling.

## 3 Preliminaries

In this section, we introduce the mathematical formulation of flow matching and describe how the denoising process can be mapped as a multi-step MDP.

**Flow Matching.** Let $\boldsymbol{x}_0 \sim X_0$ be a data sample from the true distribution, and $\boldsymbol{x}_1 \sim X_1$ denote a noise sample. Recent advanced image-generation models (e.g., [4, 5]) and video-generation models (e.g., [24, 26, 25, 27]) adopt the Rectified Flow [3] framework, which defines the "noised" data $\boldsymbol{x}_t$ as

$$\boldsymbol{x}_t = (1-t)\,\boldsymbol{x}_0 \, + \, t\,\boldsymbol{x}_1, \tag{1}$$

for $t \in [0,1]$. Then a transformer model are trained to directly regress the velocity field $\boldsymbol{v}_\theta(\boldsymbol{x}_t, t)$ by minimizing the Flow Matching objective [2, 3]:

$$\mathcal{L}(\theta) = \mathbb{E}_{t,\,\boldsymbol{x}_0 \sim X_0,\,\boldsymbol{x}_1 \sim X_1}\big[\,\|\boldsymbol{v} \, - \, \boldsymbol{v}_\theta(\boldsymbol{x}_t, t)\|^2\big], \tag{2}$$

where the target velocity field is $\boldsymbol{v} = \boldsymbol{x}_1 - \boldsymbol{x}_0$.

**Denoising as an MDP.** As shown in [12], the iterative denoising process in flow matching models can be formulated as a Markov Decision Process (MDP) $(\mathcal{S}, \mathcal{A}, \rho_0, P, R)$. The state at step $t$ is $\boldsymbol{s}_t \triangleq (\boldsymbol{c}, t, \boldsymbol{x}_t)$, the action is the denoised sample $\boldsymbol{a}_t \triangleq \boldsymbol{x}_{t-1}$ predicted by the model, and the policy is $\pi(\boldsymbol{a}_t \mid \boldsymbol{s}_t) \triangleq p_\theta(\boldsymbol{x}_{t-1} \mid \boldsymbol{x}_t, \boldsymbol{c})$. The transition is deterministic: $P(\boldsymbol{s}_{t+1} \mid \boldsymbol{s}_t, \boldsymbol{a}_t) \triangleq (\delta_{\boldsymbol{c}}, \delta_{t-1}, \delta_{\boldsymbol{x}_{t-1}})$, and the initial state distribution is $\rho_0(\boldsymbol{s}_0) \triangleq (p(\boldsymbol{c}), \delta_T, \mathcal{N}(\boldsymbol{0}, \mathbf{I}))$, where $\delta_y$ is the Dirac delta distribution centered at $y$. The reward is only given at the final step: $R(\boldsymbol{s}_t, \boldsymbol{a}_t) \triangleq r(\boldsymbol{x}_0, \boldsymbol{c})$ if $t = 0$, and 0 otherwise.

## 4 Flow-GRPO

In this section, we present Flow-GRPO, which enhances flow models using online RL. We begin by revisiting the core idea of GRPO [16] and adapting it to flow matching. We then show how to convert

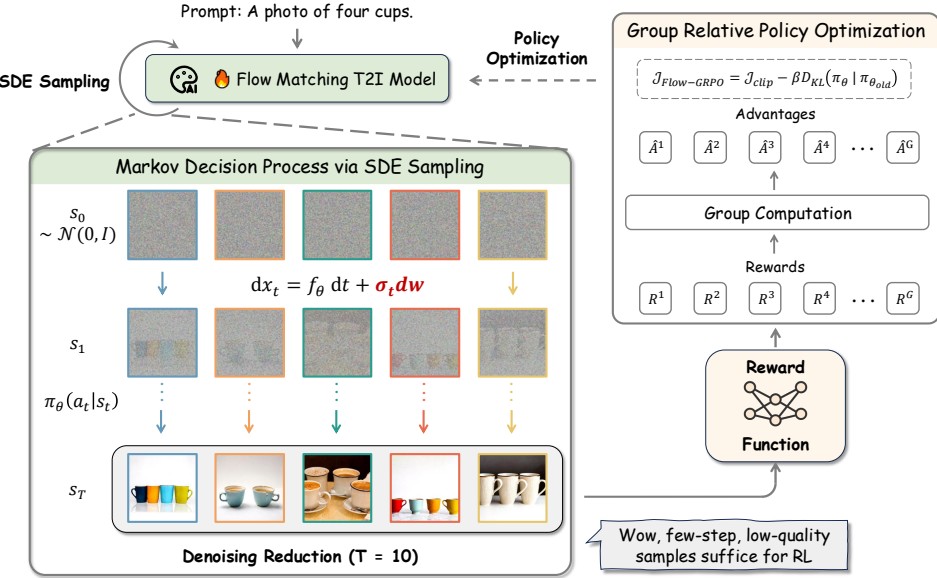

Figure 2: **Overview of Flow-GRPO.** Given a prompt set, we introduce an ODE-to-SDE strategy to enable stochastic sampling for online RL. With Denoising Reduction (only T = 10 steps), we efficiently gather low-quality but still informative trajectories. Rewards from these trajectories feed the GRPO loss, which updates the model online and yields an aligned policy.

the deterministic ODE sampler into a SDE sampler with the same marginal distribution, introducing the stochasticity needed for applying GRPO. Finally, we introduce Denoise Reduction, a practical sampling strategy that significantly speeds up training without sacrificing performance.

**GRPO on Flow Matching.** RL aims to learn a policy that maximizes the expected cumulative reward. This is often formulated as optimizing a policy $\pi_\theta$ with a regularized objective:

$$\max_\theta \mathbb{E}_{(s_0, a_0, \ldots, s_T, a_T) \sim \pi_\theta} \left[ \sum_{t=0}^{T} \left( R(s_t, a_t) - \beta D_{\mathrm{KL}}(\pi_\theta(\cdot \mid s_t) \| \pi_{\mathrm{ref}}(\cdot \mid s_t)) \right) \right]. \tag{3}$$

Unlike other policy based methods like PPO [20], GRPO [16] provides a lightweight alternative, which introduces a group relative formulation to estimate the advantage.

Recall that the denoising process can be formulated as an MDP, as shown in Section 3. Given a prompt $c$, the flow model $p_\theta$ samples a group of $G$ individual images $\{x_0^i\}_{i=1}^G$ and the corresponding reverse-time trajectories $\{(x_T^i, x_{T-1}^i, \cdots, x_0^i)\}_{i=1}^G$. Then, the advantage of the $i$-th image is calculated by normalizing the group-level rewards as follows:

$$\hat{A}_t^i = \frac{R(x_0^i, c) - \mathrm{mean}(\{R(x_0^i, c)\}_{i=1}^G)}{\mathrm{std}(\{R(x_0^i, c)\}_{i=1}^G)}. \tag{4}$$

GRPO optimizes the policy model by maximizing the following objective:

$$\mathcal{J}_{\text{Flow-GRPO}}(\theta) = \mathbb{E}_{c \sim \mathcal{C}, \{x^i\}_{i=1}^G \sim \pi_{\theta_{\mathrm{old}}}(\cdot | c)} f(r, \hat{A}, \theta, \varepsilon, \beta), \tag{5}$$

where

$$f(r, \hat{A}, \theta, \varepsilon, \beta) = \frac{1}{G} \sum_{i=1}^G \frac{1}{T} \sum_{t=0}^{T-1} \left( \min \left( r_t^i(\theta) \hat{A}_t^i, \ \mathrm{clip}\left( r_t^i(\theta), 1 - \varepsilon, 1 + \varepsilon \right) \hat{A}_t^i \right) - \beta D_{\mathrm{KL}}(\pi_\theta \| \pi_{\mathrm{ref}}) \right),$$

$$r_t^i(\theta) = \frac{p_\theta(x_{t-1}^i \mid x_t^i, c)}{p_{\theta_{\mathrm{old}}}(x_{t-1}^i \mid x_t^i, c)}.$$

**From ODE to SDE.** GRPO relies on stochastic sampling in Eq. 4 and Eq. 5 to generate diverse trajectories for advantage estimation and exploration. Diffusion models naturally support this: the

forward process adds Gaussian noise step by step, and the reverse process approximates a score-based SDE solver via a Markov chain with decreasing variance. In contrast, flow matching models use a deterministic ODE for the forward process:

$$\mathrm{d}\boldsymbol{x}_t = \boldsymbol{v}_t \mathrm{d}t, \tag{6}$$

where $\boldsymbol{v}_t$ is learned via the flow matching objective in Eq. 2. A common sampling method is to discretize this ODE, yielding a one-to-one mapping between successive time steps.

This deterministic approach fails to meet the GRPO policy update requirements in two key ways: (1) $r_t^i(\theta)$ in Eq. 5 requires computing $p(\boldsymbol{x}_{t-1} \mid \boldsymbol{x}_t, \boldsymbol{c})$, which becomes computationally expensive under deterministic dynamics due to divergence estimation. (2) More importantly, RL depends on exploration. As shown in Section 5.3, reduced randomness greatly lowers training efficiency. Deterministic sampling, with no randomness beyond the initial seed, is especially problematic.

To address this limitation, we convert the deterministic Flow-ODE from Eq. 6 into an equivalent SDE that matches the original model's marginal probability density function at all timesteps. We outline the key process here. A detailed proof is provided in Appendix A. Following [23, 28, 29], we construct a reverse-time SDE formulation that preserves the marginal distribution:

$$\mathrm{d}\boldsymbol{x}_t = \left( \boldsymbol{v}_t(\boldsymbol{x}_t) - \frac{\sigma_t^2}{2} \nabla \log p_t(\boldsymbol{x}_t) \right) \mathrm{d}t + \sigma_t \mathrm{d}\boldsymbol{w}, \tag{7}$$

where $\mathrm{d}\boldsymbol{w}$ denotes Wiener process increments and $\sigma_t$ control the level of stachasticity during generation. For rectified flow, Eq. 7 is specified as:

$$\mathrm{d}\boldsymbol{x}_t = \left[ \boldsymbol{v}_t(\boldsymbol{x}_t) + \frac{\sigma_t^2}{2t} \left( \boldsymbol{x}_t + (1-t)\boldsymbol{v}_t(\boldsymbol{x}_t) \right) \right] \mathrm{d}t + \sigma_t \mathrm{d}\boldsymbol{w}. \tag{8}$$

Applying Euler-Maruyama discretization yields the final update rule:

$$\boxed{\boldsymbol{x}_{t+\Delta t} = \boldsymbol{x}_t + \left[ \boldsymbol{v}_\theta(\boldsymbol{x}_t, t) + \frac{\sigma_t^2}{2t} \left( \boldsymbol{x}_t + (1-t)\boldsymbol{v}_\theta(\boldsymbol{x}_t, t) \right) \right] \Delta t + \sigma_t \sqrt{\Delta t}\, \epsilon} \tag{9}$$

where $\epsilon \sim \mathcal{N}(0, \boldsymbol{I})$ injects stochasticity. We use $\sigma_t = a\sqrt{\frac{t}{1-t}}$ in this paper, where $a$ is a scalar hyper-parameter that controls the noise level (See Section 5.3 for its impact on performance).

Eq. 9 reveals that the policy $\pi_\theta(\boldsymbol{x}_{t-1} \mid \boldsymbol{x}_t, \boldsymbol{c})$ is an isotropic Gaussian distribution. We can easily compute the KL divergence between $\pi_\theta$ and the reference policy $\pi_{\mathrm{ref}}$ in Eq. 5 as a closed form:

$$D_{\mathrm{KL}}(\pi_\theta || \pi_{\mathrm{ref}}) = \frac{\|\overline{\boldsymbol{x}}_{t+\Delta t, \theta} - \overline{\boldsymbol{x}}_{t+\Delta t, \mathrm{ref}}\|^2}{2\sigma_t^2 \Delta t} = \frac{\Delta t}{2} \left( \frac{\sigma_t(1-t)}{2t} + \frac{1}{\sigma_t} \right)^2 \|\boldsymbol{v}_\theta(\boldsymbol{x}_t, t) - \boldsymbol{v}_{\mathrm{ref}}(\boldsymbol{x}_t, t)\|^2$$

**Denoising Reduction.** To produce high-quality images, flow models typically require many denoising steps, making data collection costly for online RL. However, we find that large timesteps are unnecessary during online RL training. We can use significantly fewer denoising steps during sample generation, while retaining the original denoising steps during inference to get high-quality samples. Note that we set the timestep $T$ as 10 in training, while the inference timestep $T$ is set as the original default setting ($T = 40$) for SD3.5-M. Our experiments reveals that this approach enables fast training without sacrificing image quality at test time.

# 5 Experiments

This section empirically evaluates Flow-GRPO's ability to improve flow matching models on three tasks. (1) Composition Image Generation: This task requires precise object arrangement and attribute control. We report the results on GenEval. (2) Visual Text Rendering: a rule-based task that evaluates the accurate rendering of the text specified in the prompt. (3) Human Preference Alignment: This task aims to align T2I models with human preferences.

## 5.1 Experimental Setup

We introduce three tasks, detailing their respective prompts and reward definitions. For hyperparameter details and compute resource specifications, please refer to Appendix B.3 and Appendix B.4.

**Compositional Image Generation.** GenEval [17] assesses T2I models on complex compositional prompts—like object counting, spatial relations, and attribute binding—across six difficult compositional image generation tasks. We use its official evaluation pipeline, which detects object bounding boxes and colors, then infers their spatial relations. Training prompts are generated using official GenEval scripts, which apply templates and random combinations to construct the prompt dataset. The test set is strictly deduplicated: prompts differing only in object order (e.g., "a photo of A and B" vs. "a photo of B and A") are treated as identical, and these variants are removed from the training set. Based on the base model's initial accuracy across the six tasks, we set the prompt ratio as $\mathrm{Position : Counting : Attribute\ Binding : Colors : Two\ Objects : Single\ Object} = 7 : 5 : 3 : 1 : 1 : 0$. Rewards are rule-based: (1) **Counting:** $r = 1 - |N_{\mathrm{gen}} - N_{\mathrm{ref}}|/N_{\mathrm{ref}}$; (2) **Position / Color:** If the object count is correct, a partial reward is assigned; the remainder is granted when the predicted position or color is also correct.

**Visual Text Rendering [8].** Text is common in images such as posters, book covers, and memes, so the ability to place accurate and coherent text inside the generated images is crucial for T2I models. In our settings, we define an text rendering task, where each prompt follows the template ''A sign that says ''text''. Specifically, the placeholder ''text'' is the exact string that should appear in the image. We use GPT4o to produce 20K training prompts and 1K test prompts. Following [58], we measure text fidelity with the reward $r = \max(1 - N_{\mathrm{e}}/N_{\mathrm{ref}}, 0)$, where $N_{\mathrm{e}}$ is the minimum edit distance between the rendered text and the target text and $N_{\mathrm{ref}}$ is the number of characters inside the quotation marks in the prompt. This reward also serves as our metric of text accuracy.

**Human Preference Alignment [19].** This task aims to align T2I models with human preferences. We use PickScore [19] as our reward model, which is based on large-scale human annotated pairwise comparisons of images generated from the same prompt. For each image and prompt pair, PickScore provides an overall score that evaluates multiple criteria, such as the alignment of the image with the prompt and its visual quality.

**Image Quality Evaluation Metric.** Since the T2I model is trained to maximize a predefined reward, it is vulnerable to reward hacking, where the reward increases but image quality or diversity declines. This study aims to make online RL effective for T2I generation without noticeably compromising quality or diversity. To detect reward hacking beyond task-specific accuracy, we evaluate four automatic image quality metrics: Aesthetic Score [59], DeQA [60], ImageReward [32], and UnifiedReward [61] (see Appendix B.1 for details). All metrics are computed on DrawBench [1], a comprehensive benchmark with diverse prompts for T2I models.

## 5.2 Main Results

Figure 1 and Table 1 show Flow-GRPO's GenEval performance steadily improving during training, ultimately outperforming GPT-4o. This occurs while maintaining both image quality metrics and preference scores on DrawBench, a benchmark with diverse and comprehensive prompts for evaluating general model capabilities. Figure 3 offers qualitative comparisons. Beyond Compositional Image Generation, Table 2 details evaluations on Visual Text Rendering and Human Preference tasks. Flow-GRPO improved text rendering ability, again without decreasing image quality metrics and preference scores on DrawBench. See Figures 13, 14 & 15 in Appendix C.6 for related qualitative examples. For the Human Preference task, image quality did not decrease without KL regularization. However, we found that omitting KL caused a collapse in visual diversity, a form of reward hacking discussed further in Section 5.3. These results demonstrate that Flow-GRPO boosts desired capabilities while causing very little degradation to image quality or visual diversity.

**Flow-GRPO vs. Other Alignment Methods.** We compare Flow-GRPO with several alignment methods: supervised fine-tuning (SFT), Flow-DPO [14, 39], and their online variants. Flow-GRPO consistently outperforms all baselines by a significant margin. At each step, we generate a group of images using the same group size as in Flow-GRPO. The only difference lies in the update rule:

- **SFT**: Select the highest-reward image in each group and fine-tune on it.
- **Flow-DPO**: Use the highest-reward image in each group as the chosen sample and the lowest as the rejected, then apply the DPO loss.

Table 1: **GenEval Result.** Best scores are in `blue`, second-best in `green`. Results for models other than SD3.5-M are from [7] or their original papers. Obj.: Object; Attr.: Attribution.

| Model | Overall | Single Obj. | Two Obj. | Counting | Colors | Position | Attr. Binding |
|---|---|---|---|---|---|---|---|
| *Diffusion Models* | | | | | | | |
| LDM [62] | 0.37 | 0.92 | 0.29 | 0.23 | 0.70 | 0.02 | 0.05 |
| SD1.5 [62] | 0.43 | 0.97 | 0.38 | 0.35 | 0.76 | 0.04 | 0.06 |
| SD2.1 [62] | 0.50 | 0.98 | 0.51 | 0.44 | 0.85 | 0.07 | 0.17 |
| SD-XL [63] | 0.55 | 0.98 | 0.74 | 0.39 | 0.85 | 0.15 | 0.23 |
| DALLE-2 [64] | 0.52 | 0.94 | 0.66 | 0.49 | 0.77 | 0.10 | 0.19 |
| DALLE-3 [65] | 0.67 | 0.96 | 0.87 | 0.47 | 0.83 | 0.43 | 0.45 |
| *Autoregressive Models* | | | | | | | |
| Show-o [66] | 0.53 | 0.95 | 0.52 | 0.49 | 0.82 | 0.11 | 0.28 |
| Emu3-Gen [67] | 0.54 | 0.98 | 0.71 | 0.34 | 0.81 | 0.17 | 0.21 |
| JanusFlow [68] | 0.63 | 0.97 | 0.59 | 0.45 | 0.83 | 0.53 | 0.42 |
| Janus-Pro-7B [69] | 0.80 | 0.99 | 0.89 | 0.59 | 0.90 | 0.79 | 0.66 |
| GPT-4o [18] | 0.84 | 0.99 | 0.92 | 0.85 | 0.92 | 0.75 | 0.61 |
| *Flow Matching Models* | | | | | | | |
| FLUX.1 Dev [5] | 0.66 | 0.98 | 0.81 | 0.74 | 0.79 | 0.22 | 0.45 |
| SD3.5-L [4] | 0.71 | 0.98 | 0.89 | 0.73 | 0.83 | 0.34 | 0.47 |
| SANA-1.5 4.8B [70] | 0.81 | 0.99 | 0.93 | 0.86 | 0.84 | 0.59 | 0.65 |
| SD3.5-M [4] | 0.63 | 0.98 | 0.78 | 0.50 | 0.81 | 0.24 | 0.52 |
| **SD3.5-M+Flow-GRPO** | 0.95 | 1.00 | 0.99 | 0.95 | 0.92 | 0.99 | 0.86 |

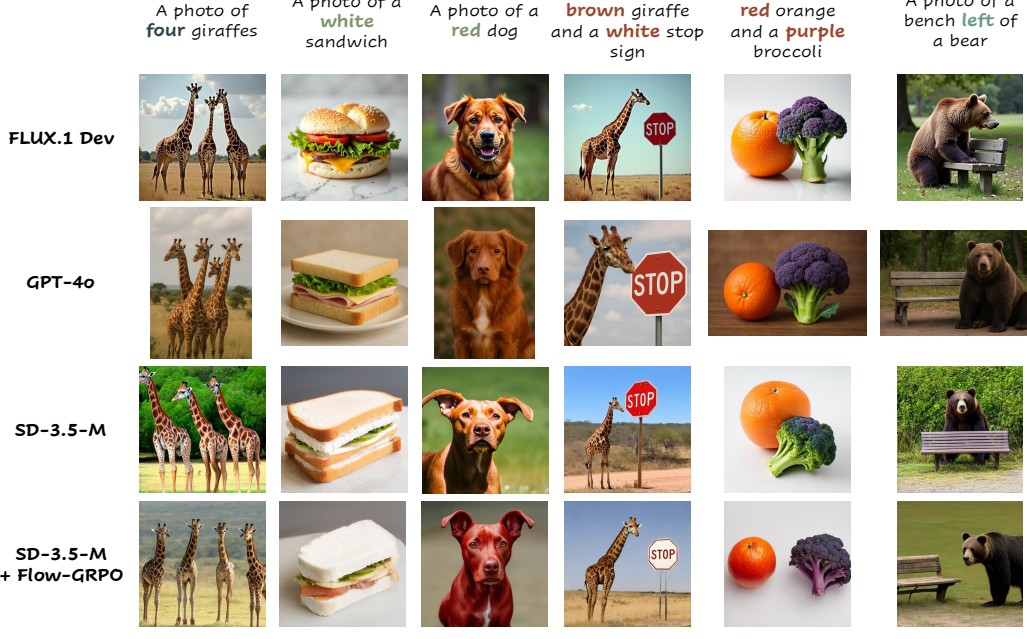

Figure 3: **Qualitative Comparison on the GenEval Benchmark.** Our approach demonstrates superior performance in **Counting**, **Colors**, **Attribute Binding**, **and** **Position**.

Offline variants use a fixed pretrained model for data collection, while online variants update their data collection models every 40 steps. As shown in Figure 4, Flow-GRPO outperforms all baselines. Online DPO also surpasses its offline counterpart, consistent with [15]. For the second-best online DPO, a hyperparameter search on its key parameter $\beta$ revealed that smaller values are not always optimal; excessively small $\beta$ values can cause training collapse. Appendix C presents more comprehensive comparisons covering additional methods and tasks.

Table 2: **Performance on Compositional Image Generation, Visual Text Rendering, and Human Preference** benchmarks, evaluated by task performance on test prompts, and by image quality and preference scores on DrawBench prompts. ImgRwd: ImageReward; UniRwd: UnifiedReward.

| Model | Task Metric | | | Image Quality | | Preference Score | | |
|---|---|---|---|---|---|---|---|---|
| | GenEval | OCR Acc. | PickScore | Aesthetic | DeQA | ImgRwd | PickScore | UniRwd |
| SD3.5-M | 0.63 | 0.59 | 21.72 | 5.39 | 4.07 | 0.87 | 22.34 | 3.33 |
| *Compositional Image Generation* | | | | | | | | |
| Flow-GRPO (w/o KL) | 0.95 | — | — | 4.93 | 2.77 | 0.44 | 21.16 | 2.94 |
| Flow-GRPO (w/ KL) | 0.95 | — | — | 5.25 | 4.01 | 1.03 | 22.37 | 3.51 |
| *Visual Text Rendering* | | | | | | | | |
| Flow-GRPO (w/o KL) | — | 0.93 | — | 5.13 | 3.66 | 0.58 | 21.79 | 3.15 |
| Flow-GRPO (w/ KL) | — | 0.92 | — | 5.32 | 4.06 | 0.95 | 22.44 | 3.42 |
| *Human Preference Alignment* | | | | | | | | |
| Flow-GRPO (w/o KL) | — | — | 23.41 | 6.15 | 4.16 | 1.24 | 23.56 | 3.57 |
| Flow-GRPO (w/ KL) | — | — | 23.31 | 5.92 | 4.22 | 1.28 | 23.53 | 3.66 |

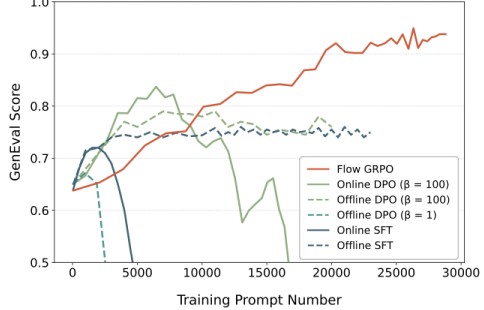

Figure 4: Comparison with Other Alignment Methods on the Compositional Generation Task.

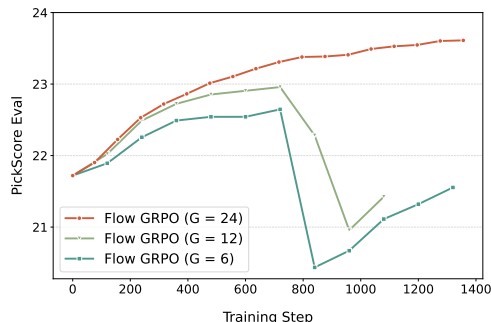

Figure 5: Ablation Studies on Different Group Size $G$. Higher group size performs better.

## 5.3 Analysis

This section presents several analyses to better understand the behavior and robustness of Flow-GRPO. We examine issues such as reward hacking, the impact of denoising reduction and noise levels, the effect of group size, and the model's generalization ability. We provide additional analyses in the Appendix C.

**Reward Hacking.** We use KL regularization to mitigate reward hacking by tuning the KL coefficient to keep the divergence small and nearly constant during training, keeping the model close to its pretrained weights. This allows task-specific reward optimization without harming overall performance. As shown in Table 2, removing the KL constraint for Compositional Image Generation and Visual Text Rendering significantly reduces image quality and preference scores on DrawBench. In contrast, a properly tuned KL preserves quality while achieving similar gains on task-specific metrics. In the Human Preference Alignment task, removing KL does not affect image quality, likely due to overlap between PickScore and evaluation metrics, but causes a collapse in visual diversity. Outputs converge to a single style, with different seeds producing nearly identical results. KL regularization prevents this collapse and maintains diversity. See Figure 12 in Appendix C.5 for training curves and Figure 6 for more examples.

**Effect of Denoising Reduction.** Figure 7 (a) highlights Denoising Reduction's significant impact on accelerating training. To explore how different timesteps affect optimization, these experiments are conducted without the KL constraint. Reducing data collection timesteps from 40 to 10 achieves over a $4\times$ speedup across all three tasks, without impacting final reward. Further reducing to 5 does not consistently improve speed and sometimes slows training, so we choose 10 timesteps for later

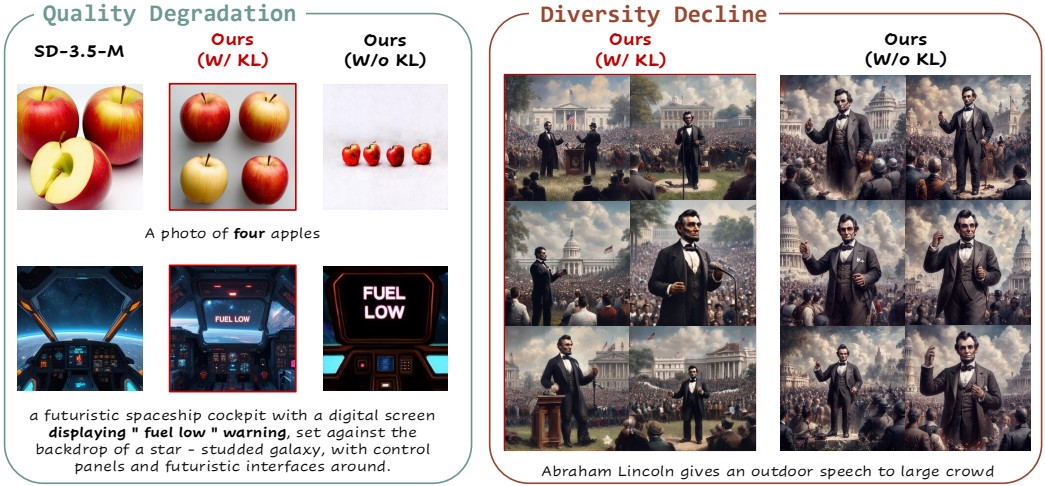

Figure 6: **Effect of KL Regularization.** The KL penalty effectively suppresses reward hacking, preventing **Quality Degradation** (for GenEval and OCR) and **Diversity Decline** (for PickScore).

experiments. For the other two tasks, learning curves of reward versus training time are presented in Figure 9 in the Appendix C.2.

**Effect of Noise Level.** Higher $\sigma_t$ in the SDE boosts image diversity and exploration, vital for RL training. We control this exploration with a noise level $a$ (Eq. 9). Figure 7 (b) shows the impact of $a$ on performance. A small $a$ (e.g., 0.1) limits exploration and slows reward improvement. Increasing $a$ (up to 0.7) boosts exploration and speeds up reward gains. Beyond this point (e.g., from 0.7 to 1.0), further increases provide no additional benefit, as exploration is already sufficient. We also observe that injecting too much noise by further increasing $a$ degrades image quality, resulting in zero reward and failed training.

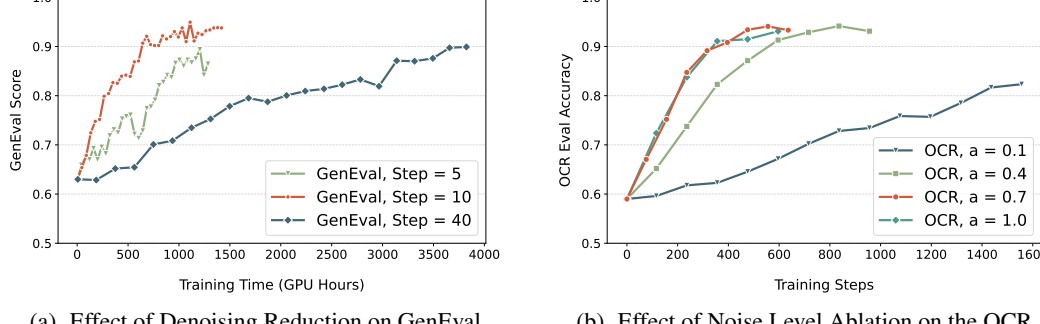

(a). Effect of Denoising Reduction on GenEval          (b). Effect of Noise Level Ablation on the OCR

Figure 7: Ablation studies on our critical design choices. **(a) Denoising Reduction**: Fewer denoising steps accelerate convergence and yield similar performance. **(b) Noise Level:** Moderate noise level ($a = 0.7$) maximises OCR accuracy, while too little noise hampers exploration.

**Effect of Group Size.** Figure 5 shows the effect of group size $G$ using PickScore as the reward function. When the group size was reduced to $G = 12$ and $G = 6$, training became unstable and eventually collapsed, whereas $G = 24$ remained stable throughout the process. We observe that smaller group sizes produce inaccurate advantage estimates, increasing variance and leading to training collapse, a phenomenon also reported in [71, 72].

**Generalization Analysis.** Flow-GRPO demonstrates strong generalization on unseen scenarios from GenEval (Table 4). Specifically, it captures object number, color, and spatial relations, generalizing well to unseen object classes. It also effectively controls object count, generalizing from training on $2 - 4$ objects to generate $5 - 6$ or $12$ objects. Furthermore, Table 3 shows Flow-GRPO achieves

significant gains on T2I-CompBench++ [6, 73]. This comprehensive benchmark for open-world compositional T2I generation features object classes and relationships substantially different from our model's GenEval-style training data.

Table 3: **T2I-CompBench++ Result.** This evaluation uses the same model presented in Table 1, which was trained on the GenEval-generated dataset. The best score is in blue .

| Model | Color | Shape | Texture | 2D-Spatial | 3D-Spatial | Numeracy | Non-Spatial |
|---|---|---|---|---|---|---|---|
| Janus-Pro-7B [69] | 0.5145 | 0.3323 | 0.4069 | 0.1566 | 0.2753 | 0.4406 | 0.3137 |
| EMU3 [67] | 0.7913 | 0.5846 | 0.7422 | — | — | — | — |
| FLUX.1 Dev [5] | 0.7407 | 0.5718 | 0.6922 | 0.2863 | 0.3866 | 0.6185 | 0.3127 |
| SD3.5-M [4] | 0.7994 | 0.5669 | 0.7338 | 0.2850 | 0.3739 | 0.5927 | 0.3146 |
| **SD3.5-M+Flow-GRPO** | 0.8379 | 0.6130 | 0.7236 | 0.5447 | 0.4471 | 0.6752 | 0.3195 |

Table 4: **Flow-GRPO demonstrates strong generalization.** Unseen Objects: Trained on 60 object classes, evaluated on 20 unseen classes. Unseen Counting: Trained to render 2, 3, or 4 objects, and evaluated in two settings: rendering 5 or 6 objects, and rendering 12 objects.

| Method | Unseen Objects | | | | | | | Unseen Counting | |
|---|---|---|---|---|---|---|---|---|---|
| | Overall | Single Obj. | Two Obj. | Counting | Colors | Position | Attr. Binding | 5−6 Objects | 12 Objects |
| SD3.5-M | 0.64 | 0.96 | 0.73 | 0.53 | 0.87 | 0.26 | 0.47 | 0.13 | 0.02 |
| **SD3.5-M+Flow-GRPO** | 0.90 | 1.00 | 0.94 | 0.86 | 0.97 | 0.84 | 0.77 | 0.48 | 0.12 |

## 6  Conclusion

We have presented Flow-GRPO, the first method to integrate online policy gradient RL into flow matching models. By converting deterministic ODEs to SDEs and reducing denoising steps during training, Flow-GRPO enables efficient RL-based optimization without noticeably compromising image quality or diversity. Our method significantly improves performance on compositional generation, text rendering, and human preference alignment, with minimal reward hacking. Flow-GRPO offers a simple and general framework for applying online RL to flow-based generative models.

**Limitations & Future Work.**  Although this work focuses on T2I tasks, Flow-GRPO has potential for video generation [25, 27], raising several future directions: (1) Reward Design: Simple heuristics, such as using object detectors or trackers as rule-based rewards, can encourage physical realism and temporal consistency, but more advanced reward models are needed. (2) Balancing Multiple Rewards: Video generation requires optimizing multiple objectives, including realism, smoothness, and coherence. Balancing these competing goals remains challenging and demands careful tuning. (3) Scalability: Video generation is far more resource-intensive than T2I, so applying Flow-GRPO at scale requires more efficient data collection and training pipelines. Additionally, better methods for preventing reward hacking are worth exploring. While KL regularization helps significantly, it requires longer training and occasional reward hacking occurs for certain prompts.

## Acknowledgements

This work was partially supported by the JC STEM Lab of AI for Science and Engineering, funded by The Hong Kong Jockey Club Charities Trust, the Research Grants Council of Hong Kong (Project No. CUHK14213224). We gratefully acknowledge Mingwu Zheng for his insightful discussions on the proof and Zhanhui Zhou for his valuable comments that improved the clarity of this paper.

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

# Appendix of Flow-GRPO:
# Training Flow Matching Models via Online RL

Our Appendix consists of 4 sections. Readers can click on each section number to navigate to the corresponding section:

- Section A provides detailed derivations of stochastic sampling in flow matching models.
- Section B presents details about our experimental setup.
- Section C offers some additional experimental results, including 1) the comparison with other alignment methods, 2) ablation of denoising reduction on OCR accuracy and pickscore, 3) ablation of initial noise, 4) additional results on FLUX.1-Dev, 5) the learning curves of Flow-GRPO on three tasks, 6) additional qualitative results, and 7) evolution of evaluation images during training.
- Section D provides a visualization of training samples under the denoising reduction strategy.

In addition to this Appendix, we also provide more visualization results, see this website. We encourage the readers to consult this HTML page for a more intuitive assessment of the improvements brought by Flow-GRPO.

# A  Mathematical Derivations for Stochastic Sampling using Flow Models

We present a detailed proof here. To compute $p_\theta(\boldsymbol{x}_{t-1} \mid \boldsymbol{x}_t, \boldsymbol{c})$ in Equation 5 during forward sampling, we adapt flow models to a stochastic differential equation (SDE). While flow models normally follow a deterministic ODE:

$$\mathrm{d}\boldsymbol{x}_t = \boldsymbol{v}_t \mathrm{d}t \tag{10}$$

We consider its stochastic counterpart. Inspired by the derivation from SDE to its probability flow ODE in SGMs [23], we aim to construct a forward SDE with specific drift and diffusion coefficients so that its marginal distribution matches that of Eq. 10. We begin with the generic form of SDE:

$$\mathrm{d}\boldsymbol{x}_t = f_{\mathrm{SDE}}(\boldsymbol{x}_t, t)\mathrm{d}t + \sigma_t \mathrm{d}\boldsymbol{w}, \tag{11}$$

Its marginal probability density $p_t(\boldsymbol{x})$ evolves according to the Fokker–Planck equation [74], i.e.,

$$\partial_t p_t(x) = -\nabla \cdot [f_{\mathrm{SDE}}(\boldsymbol{x}_t, t) p_t(\boldsymbol{x})] + \frac{1}{2}\nabla^2[\sigma_t^2 p_t(\boldsymbol{x})] \tag{12}$$

Similarly, the marginal probability density associated with Eq. 10 evolves:

$$\partial_t p_t(\boldsymbol{x}) = -\nabla \cdot [\boldsymbol{v}_t(\boldsymbol{x}_t, t) p_t(\boldsymbol{x})] \tag{13}$$

To ensure that the stochastic process shares the same marginal distribution as the ODE, we impose:

$$-\nabla \cdot [f_{\mathrm{SDE}}\, p_t(\boldsymbol{x})] + \frac{1}{2}\nabla^2[\sigma_t^2 p_t(\boldsymbol{x})] = -\nabla \cdot [\boldsymbol{v}_t(\boldsymbol{x}_t, t) p_t(\boldsymbol{x})] \tag{14}$$

Observing that

$$\begin{aligned}
\nabla^2[\sigma_t^2 p_t(\boldsymbol{x})] &= \sigma_t^2 \nabla^2 p_t(\boldsymbol{x}) \\
&= \sigma_t^2 \nabla \cdot (\nabla p_t(\boldsymbol{x})) \\
&= \sigma_t^2 \nabla \cdot (p_t(\boldsymbol{x}) \nabla \log p_t(\boldsymbol{x}))
\end{aligned} \tag{15}$$

Substituting Eq. 15 to Eq. 14, we arrive at the drift coefficients of the target forward SDE:

$$f_{\mathrm{SDE}} = \boldsymbol{v}_t(\boldsymbol{x}_t, t) + \frac{\sigma_t^2}{2}\nabla \log p_t(\boldsymbol{x}) \tag{16}$$

Hence, we can rewrite the forward SDE in Eq. 11 as:

$$\mathrm{d}\boldsymbol{x}_t = \left( \boldsymbol{v}_t(\boldsymbol{x}_t) + \frac{\sigma_t^2}{2}\nabla \log p_t(\boldsymbol{x}_t) \right)\mathrm{d}t + \sigma_t \mathrm{d}\boldsymbol{w}, \tag{17}$$

where $\mathrm{d}\boldsymbol{w}$ denotes Wiener process increments, and $\sigma_t$ is the diffusion coefficient controlling the level of stochasticity during sampling.

The relationship between forward and reverse-time SDEs has been established in [75, 23]. Specifically, if the forward SDE takes the form

$$\mathrm{d}\boldsymbol{x}_t = f(\boldsymbol{x}_t, t)\, \mathrm{d}t + g(t)\, \mathrm{d}\boldsymbol{w}, \tag{18}$$

then the corresponding reverse-time SDE is

$$\mathrm{d}\boldsymbol{x}_t = \left[ f(\boldsymbol{x}_t, t) - g^2(t)\nabla \log p_t(\boldsymbol{x}_t) \right]\mathrm{d}t + g(t)\, \mathrm{d}\overline{\boldsymbol{w}}. \tag{19}$$

Setting $g(t) = \sigma_t$, we obtain the reverse-time SDE corresponding to Eq. 17 as

$$\mathrm{d}\boldsymbol{x}_t = \left[ \boldsymbol{v}_t(\boldsymbol{x}_t) + \frac{\sigma_t^2}{2}\nabla \log p_t(\boldsymbol{x}_t) - \sigma_t^2 \nabla \log p_t(\boldsymbol{x}_t) \right]\mathrm{d}t + \sigma_t \mathrm{d}\overline{\boldsymbol{w}}. \tag{20}$$

We thus arrive at the final form of the reverse-time SDE:

$$\mathrm{d}\boldsymbol{x}_t = \left( \boldsymbol{v}_t(\boldsymbol{x}_t) - \frac{\sigma_t^2}{2}\nabla \log p_t(\boldsymbol{x}_t) \right)\mathrm{d}t + \sigma_t \mathrm{d}\boldsymbol{w}, \tag{21}$$

Once the score function $\nabla \log p_t(\boldsymbol{x}_t)$ is available, the process can be simulated directly. For flow matching, this score is implicitly linked to the velocity field $\boldsymbol{v}_t$.

Specifically, let $\dot{\alpha}_t \equiv \partial \alpha_t / \partial t$. All expectations are over $\boldsymbol{x}_0 \sim X_0$ and $\boldsymbol{x}_1 \sim \mathcal{N}(0, \boldsymbol{I})$, where $X_0$ is the data distribution.

For the linear interpolation $\boldsymbol{x}_t = \alpha_t \boldsymbol{x}_0 + \beta_t \boldsymbol{x}_1$, we have:

$$p_{t|0}(\boldsymbol{x}_t | \boldsymbol{x}_0) = \mathcal{N}\left(\boldsymbol{x}_t \mid \alpha_t \boldsymbol{x}_0, \beta_t^2 \boldsymbol{I}\right), \tag{22}$$

yielding the conditional score:

$$\nabla \log p_{t|0}(\boldsymbol{x}_t | \boldsymbol{x}_0) = -\frac{\boldsymbol{x}_t - \alpha_t \boldsymbol{x}_0}{\beta_t^2} = -\frac{\boldsymbol{x}_1}{\beta_t}. \tag{23}$$

The marginal score becomes:

$$\nabla \log p_t(\boldsymbol{x}_t) = \mathbb{E}\left[\nabla \log p_{t|0}(\boldsymbol{x}_t | \boldsymbol{x}_0) \mid \boldsymbol{x}_t\right]$$
$$= -\frac{1}{\beta_t} \mathbb{E}[\boldsymbol{x}_1 \mid \boldsymbol{x}_t]. \tag{24}$$

For the velocity field $\boldsymbol{v}_t(\boldsymbol{x}_t)$, we derive:

$$\begin{aligned}
\boldsymbol{v}_t(\boldsymbol{x}) &= \mathbb{E}\left[\dot{\alpha}_t \boldsymbol{x}_0 + \dot{\beta}_t \boldsymbol{x}_1 \mid \boldsymbol{x}_t = \boldsymbol{x}\right] \\
&= \dot{\alpha}_t \mathbb{E}[\boldsymbol{x}_0 \mid \boldsymbol{x}_t = \boldsymbol{x}] + \dot{\beta}_t \mathbb{E}[\boldsymbol{x}_1 \mid \boldsymbol{x}_t = \boldsymbol{x}] \\
&= \dot{\alpha}_t \mathbb{E}\left[\frac{\boldsymbol{x}_t - \beta_t \boldsymbol{x}_1}{\alpha_t} \mid \boldsymbol{x}_t = \boldsymbol{x}\right] + \dot{\beta}_t \mathbb{E}[\boldsymbol{x}_1 \mid \boldsymbol{x}_t = \boldsymbol{x}] \\
&= \frac{\dot{\alpha}_t}{\alpha_t} \boldsymbol{x} - \frac{\dot{\alpha}_t \beta_t}{\alpha_t} \mathbb{E}[\boldsymbol{x}_1 \mid \boldsymbol{x}_t = \boldsymbol{x}] + \dot{\beta}_t \mathbb{E}[\boldsymbol{x}_1 \mid \boldsymbol{x}_t = \boldsymbol{x}] \\
&= \frac{\dot{\alpha}_t}{\alpha_t} \boldsymbol{x} - \left(\dot{\beta}_t \beta_t - \frac{\dot{\alpha}_t \beta_t^2}{\alpha_t}\right) \nabla \log p_t(\boldsymbol{x}),
\end{aligned} \tag{25}$$

Substituting $\alpha_t = 1 - t$ and $\beta_t = t$ simplifies Equation 25 to:

$$\boldsymbol{v}_t(\boldsymbol{x}) = -\frac{\boldsymbol{x}}{1-t} - \frac{t}{1-t} \nabla \log p_t(\boldsymbol{x}). \tag{26}$$

Solving for the score yields:

$$\nabla \log p_t(\boldsymbol{x}) = -\frac{\boldsymbol{x}}{t} - \frac{1-t}{t} \boldsymbol{v}_t(\boldsymbol{x}). \tag{27}$$

Substituting Equation 27 into 21 gives the final SDE:

$$d\boldsymbol{x}_t = \left[\boldsymbol{v}_t(\boldsymbol{x}_t) + \frac{\sigma_t^2}{2t}\left(\boldsymbol{x}_t + (1-t)\boldsymbol{v}_t(\boldsymbol{x}_t)\right)\right] dt + \sigma_t d\boldsymbol{w}. \tag{28}$$

Applying Euler-Maruyama discretization yields the update rule:

$$\boxed{\boldsymbol{x}_{t+\Delta t} = \boldsymbol{x}_t + \left[\boldsymbol{v}_\theta(\boldsymbol{x}_t, t) + \frac{\sigma_t^2}{2t}\left(\boldsymbol{x}_t + (1-t)\boldsymbol{v}_\theta(\boldsymbol{x}_t, t)\right)\right] \Delta t + \sigma_t \sqrt{\Delta t}\, \epsilon,} \tag{29}$$

where $\epsilon \sim \mathcal{N}(0, \boldsymbol{I})$ injects stochasticity.

## B  Further Details on the Experimental Setup

### B.1  Quality Metrics

The details of quality metrics are as follows:

- Aesthetic score [59]: a CLIP-based linear regressor that predicts an image's aesthetic score.
- DeQA score [60]: a multimodal large language model based image-quality assessment (IQA) model that quantifies how distortions, texture damage, and other low-level artefacts affect perceived quality.
- ImageReward [32]: a general purpose T2I human preference reward model that captures text–image alignment, visual fidelity, and harmlessness.
- UnifiedReward [61]: a recently proposed unified reward model for multimodal understanding and generation that currently achieves state-of-the-art performance on the human preference assessment leaderboard.

## B.2 Model Specification

The following table lists the base model and the reward models and their corresponding links.

| Models | Links |
|--------|-------|
| SD3.5-M [4] | https://huggingface.co/stabilityai/stable-diffusion-3.5-medium |
| Aesthetic Score [59] | https://github.com/LAION-AI/aesthetic-predictor |
| PickScore [19] | https://huggingface.co/yuvalkirstain/PickScore_v1 |
| DeQA score [60] | https://huggingface.co/zhiyuanyou/DeQA-Score-Mix3 |
| ImageReward [32] | https://huggingface.co/THUDM/ImageReward |
| UnifiedReward [61] | https://huggingface.co/CodeGoat24/UnifiedReward-7b-v1.5 |

## B.3 Hyperparameters Specification

Except for $\beta$, GRPO hyperparameters are fixed across tasks. We use a sampling timestep $T = 10$ and an evaluation timestep $T = 40$. Other settings include a group size $G = 24$, an noise level $a = 0.7$ and an image resolution of 512. The KL ratio $\beta$ is set to 0.04 for GenEval and Text Rendering, and 0.01 for Pickscore. We use Lora with $\alpha = 64$ and $r = 32$.

## B.4 Compute Resources Specification

We train our model using 24 NVIDIA A800 GPUs. The learning curves in Appendix C.5 provide details on the specific GPU hours.

# C Extended Experimental Results

## C.1 Flow-GRPO vs. Other Alignment Methods

We compare Flow-GRPO with several alignment methods: supervised fine-tuning (SFT), reward-weighted regression (Flow-RWR [14, 76]), Flow-DPO [14], and their online variants. Flow-GRPO consistently outperforms all baselines by a significant margin. At each step, we generate a group of images using the same group size as in Flow-GRPO. The only difference lies in the update rule:

- **SFT**: Select the highest-reward image in each group and fine-tune on it.
- **Flow-RWR** [14, 76]: Apply a softmax over rewards in each group and perform reward-weighted likelihood maximization.
- **Flow-DPO** [14, 39]: Use the highest-reward image in each group as the chosen sample and the lowest as the rejected, then apply the DPO loss.

Offline variants use a fixed pretrained model for data collection, while online variants update their data collection model every 40 steps. As shown in Figure 8, Flow-GRPO outperforms all other methods. The figure also indicates that DPO and SFT improve over time. In contrast, RWR does not, which aligns with experimental findings on RWR in [12]. Additionally, Online DPO surpasses offline DPO, aligning with [15]'s finding that online DPO performs better. For the second-best online DPO, a hyperparameter search on its key parameter $\beta$ revealed that smaller values are not always optimal; excessively small $\beta$ values can cause training collapse.

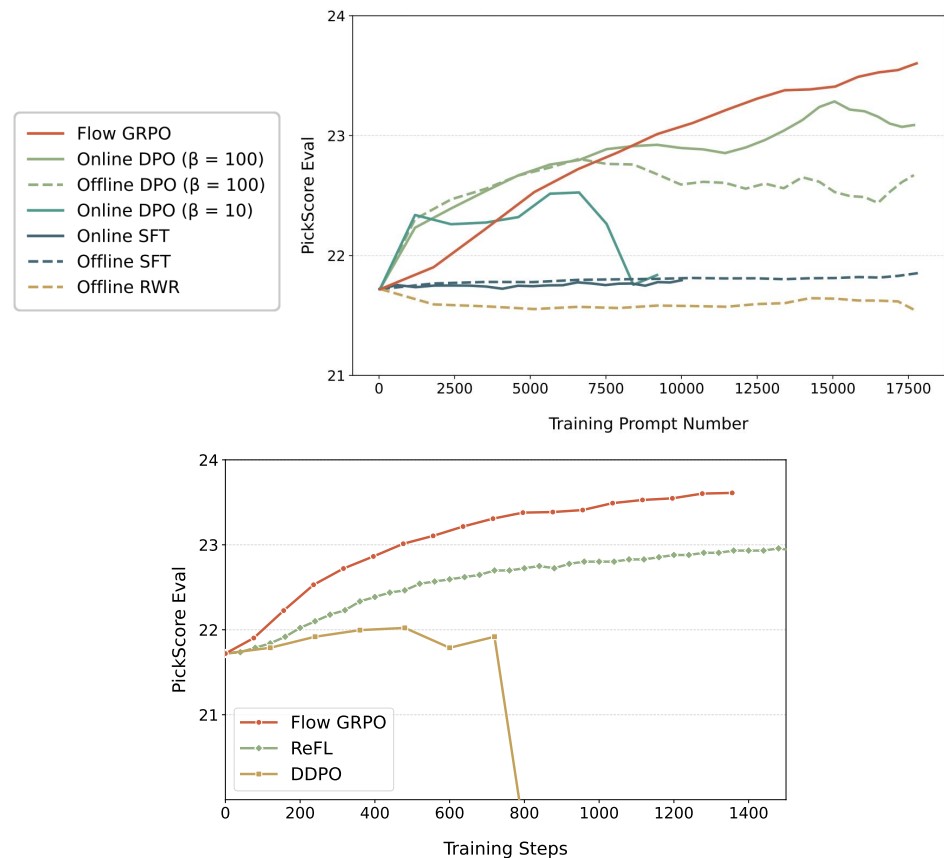

Figure 8: Comparison of Flow-GRPO and Other Alignment Methods on the Human Preference Alignment task. Since methods like DPO use different tuned batch sizes from Flow-GRPO, we use the number of training prompts on the x-axis for a fair comparison across these methods.

**DDPO.** DDPO [12] was originally developed for diffusion-based backbones, so we adapted it to flow-matching models via our ODE-to-SDE conversion. Using SD3.5-M as the base model and PickScore as the reward signal, we track the evaluation reward throughout the entire training process in Figure 8. We find that DDPO's reward increases more slowly than Flow-GRPO's and eventually collapses in the later stages, whereas Flow-GRPO trains stably and continues to improve consistently over time.

**ReFL.** ReFL [32] directly fine-tunes diffusion models by viewing reward model scores as human preference losses and back-propagating gradients to a randomly-picked late timestep $t$. Following ImageReward [32], we back-propagate gradients to a randomly chosen late timestep $t \in [30, 40]$ during denoising. Figure 8 shows that GRPO surpasses ReFL when the reward is differentiable, indicating that GRPO maintains strong performance in settings where ReFL applies. More importantly, GRPO does not require differentiable rewards, enabling direct use of state-of-the-art Vision-Language Models (VLMs) as reward providers. This offers two key advantages:

- **Sophisticated, General-Purpose Rewards:** VLMs can conduct human-like evaluations through a structured reasoning process. Given a prompt, a VLM can decompose it into key criteria, reason step by step to verify each aspect in the generated image, and then provide a comprehensive overall score. This enables a single, unified reward model to handle diverse tasks, from text-to-image generation to complex instruction-based image editing.

- **Future-Proof and Cost-Free Upgrades:** The field of VLMs is advancing at a breathtaking pace. By using a VLM as the reward source, our framework automatically benefits from these

improvements. As VLMs become more capable, the reward model becomes stronger without any additional training data or computational cost.

**ORW.**  ORW [35] is an online reward-weighted regression method that guides the model to prioritize high-reward regions. Unlike KL regularization, it employs Wasserstein-2 regularization to prevent policy collapse and maintain diversity. To ensure a fair comparison, we adopt the same experimental setup as in our Human Preference Alignment task. For ORW, we set $\beta = 0.5$ and $\alpha = 1$ (lower values led to unstable training). The steps_per_epoch parameter, which controls how frequently the data-collecting policy is updated, was chosen from $20, 40, 100, 400$ based on best performance. Table 5 reports reward scores on the test set across training steps. Following ORW's Table 1, we randomly sampled 50 DrawBench prompts and generated 64 images per prompt to compute CLIP and Diversity scores. As shown in Table 6, Flow-GRPO outperforms ORW on both metrics.

Table 5: Reward scores on the test set over training steps.

| Method | Step 0 | Step 240 | Step 480 | Step 720 | Step 960 |
|---|---|---|---|---|---|
| SD3.5-M + ORW | 28.79 | 29.05 | 29.15 | 27.58 | 23.05 |
| **SD3.5-M + Flow-GRPO** | 28.79 | **29.10** | **29.17** | **29.51** | **29.89** |

Table 6: Comparison of CLIP and diversity scores across different fine-tuning methods.

| Method | CLIP Score ↑ | Diversity Score ↑ |
|---|---|---|
| SD3.5-M | 27.99 | 0.96 |
| SD3.5-M + ORW | 28.40 | 0.97 |
| **SD3.5-M + Flow-GRPO** | **30.18** | **1.02** |

## C.2   Effect of Denoising Reduction

We show the extended Denoising Reduction ablations of Visual Text Rendering and Human Preference Alignment tasks in Figure 9.

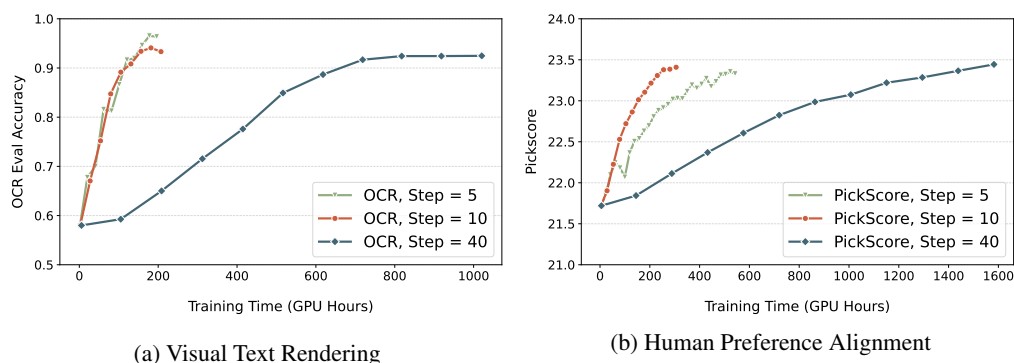

(a) Visual Text Rendering          (b) Human Preference Alignment

Figure 9: Effect of Denoising Reduction

## C.3   Effect of Initial Noise

We initialize each rollout with difference random noise to increase exploratory diversity during RL training. We perform an additioanl ablation to confirm this claim. With SD3.5-M as the base model and PickScore as the reward, we compare Flow-GRPO with different initial noise against Flow-GRPO with the same initial noise. Figure 10 shows the variant with different noise consistently achieved high rewards during the training process.

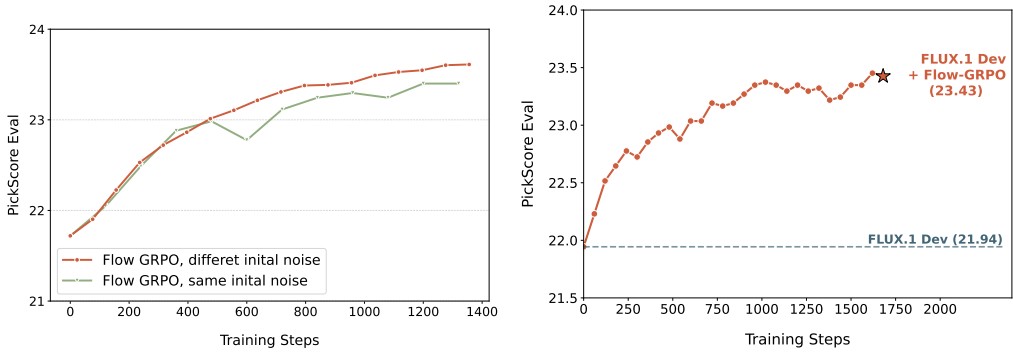

Figure 10: Effect of Initial Noise  Figure 11: Additional Results on FLUX.1-Dev

### C.4 Additional Results on FLUX.1-Dev

We run Flow-GRPO on FLUX.1-Dev [5] using PickScore as the reward signal. The reward curve rises steadily throughout training without noticeable reward hacking. Figure 11 shows the reward values over the training process, and Table 7 compares FLUX.1-Dev with FLUX.1-Dev + Flow-GRPO on DrawBench.

Table 7: Comparison of FLUX.1-Dev and Flow-GRPO fine-tuned models.

| Model | Aesthetic | DeQA | ImageReward | PickScore | UnifiedReward |
|---|---|---|---|---|---|
| FLUX.1-Dev | 5.71 | 4.31 | 0.85 | 22.62 | 3.65 |
| FLUX.1-Dev + Flow-GRPO | 6.02 | 4.24 | 1.32 | 23.97 | 3.81 |

### C.5 Learning Curves with or without KL

Figure 12 shows learning curves for three tasks, with and without KL. These results emphasize that KL regularization is not empirically equivalent to early stopping. Adding appropriate KL can achieve the same high reward as the KL-free version and maintain image quality, though it requires longer training.

### C.6 Additional Qualitative Results

Figures 13, 14 & 15 qualitatively compare SD3.5-M with its Flow-GRPO enhanced versions (with and without KL regularization) using GenEval, OCR and PickScore rewards, respectively. Flow-GRPO with KL regularization improves the target capability while maintaining image quality and minimizing reward-hacking. Conversely, removing the KL constraint significantly degrades image quality and diversity.

### C.7 Evolution of Evaluation Images During Flow-GRPO Training

To better understand the training dynamics of our proposed Flow-GRPO framework, we visualize the evolution of generated samples corresponding to fixed evaluation prompts at regular intervals during training in Figure 16, 17 & 18. For consistency, all visualizations are produced using a 40-step ODE-based sampling schedule. These qualitative results provide a visual representation of how the model progressively improves its generation quality and alignment with task objectives over time.

## D Training Sample Visualization with Denoising Reduction

In this section, we compare images obtained with SDE sampling at various steps against those produced by ODE sampling, and offer an intuitive view of the denoising reduction strategy. Figure 19 presents SD3.5-Medium samples under four inference settings: (a) ODE sampling with 40 steps; (b) SDE sampling with 40 steps; (c) SDE sampling with 10 steps; (d) SDE sampling with 5 steps.

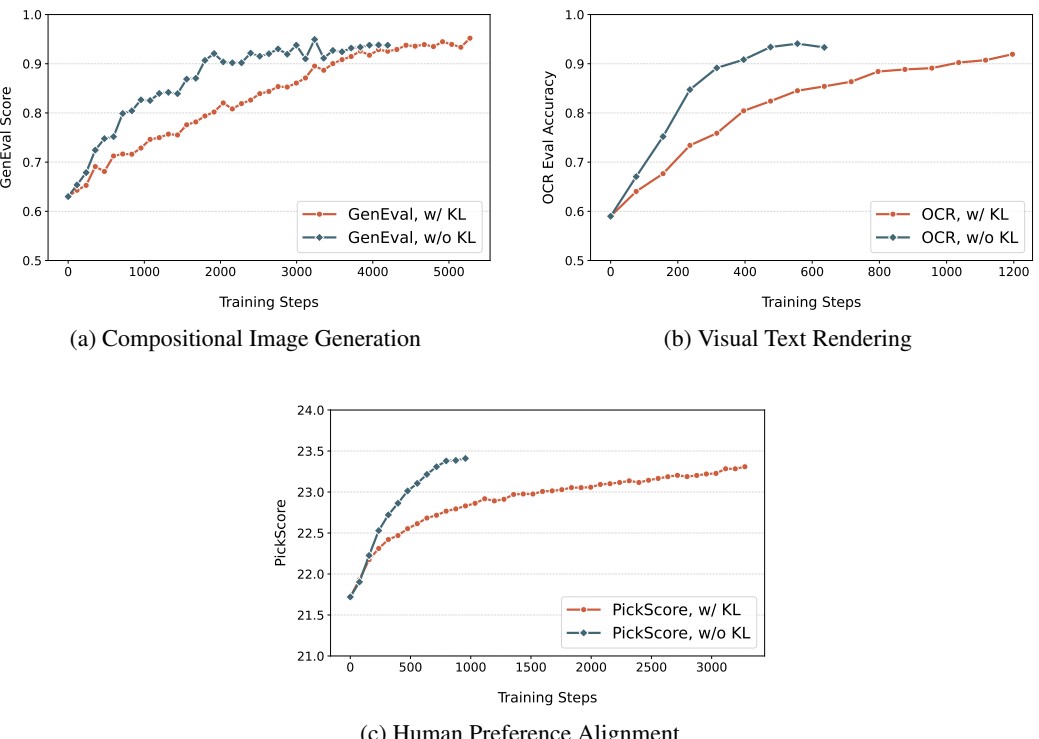

(a) Compositional Image Generation

(b) Visual Text Rendering

(c) Human Preference Alignment

Figure 12: Learning Curves with and without KL. KL penalty slows early training yet effectively suppresses reward hacking.

The 40-step ODE and SDE runs yield visually indistinguishable images, confirming that our SDE sampler preserves quality. Shortening the SDE schedule to 10 and 5 steps introduces conspicuous artifacts, like color drift and fine details blur. Contrary to expectation that such low-quality samples might hinder optimization. it actually do just the opposite and accelerate optimization. Because Flow-GRPO relies on relative preferences, it still extracts a useful reward signal, while the shorter trajectories significatly cut wall-clock time. Consequently, Flow-GRPO with denoising reduction strategy converges more quickly on both layout-oriented benchmarks such as GenEval and quality-focused metrics such as PickScore, without sacrificing final performance.

SD-3.5-M                    **Flow-GRPO**              Flow-GRPO(w/o KL)

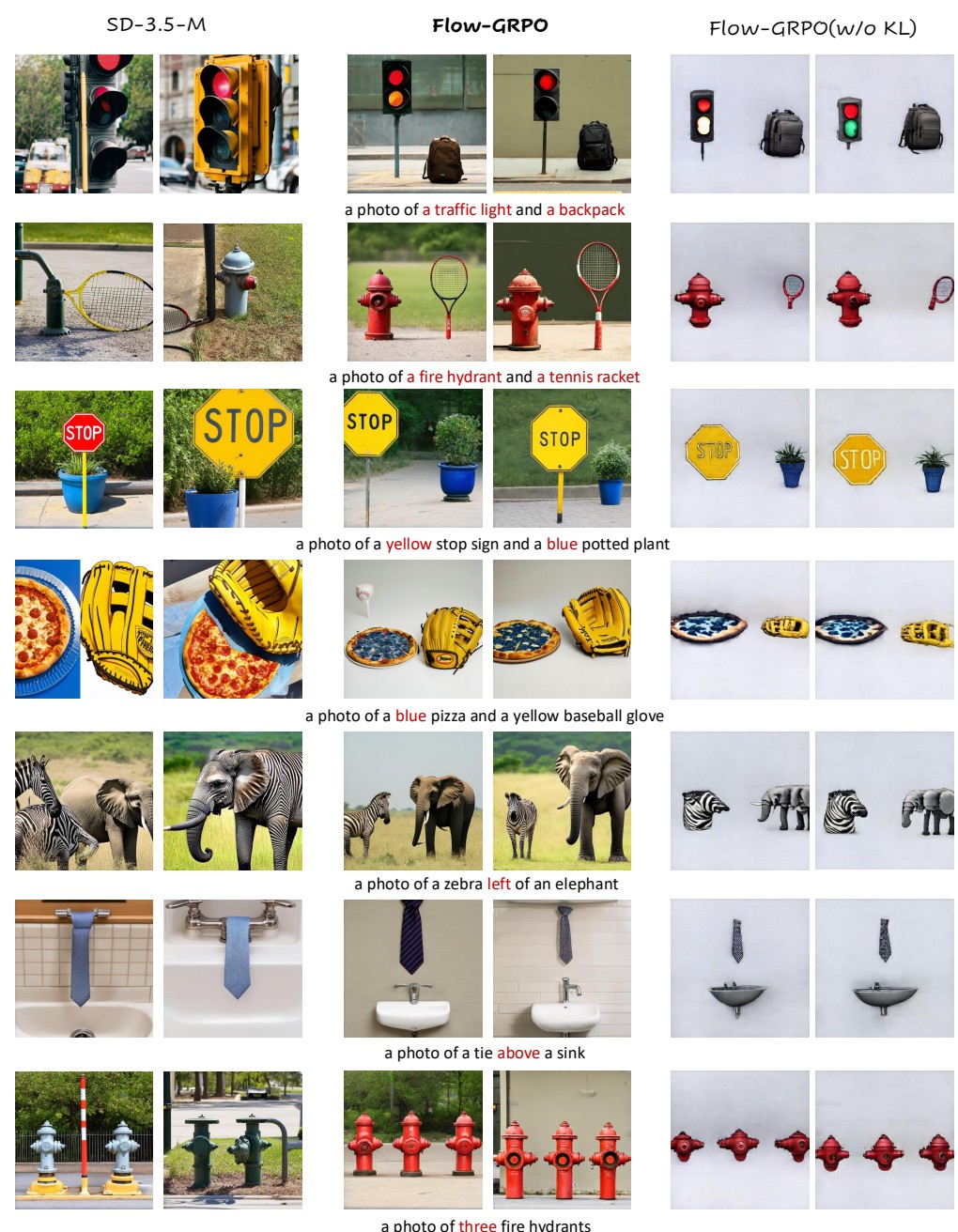

Figure 13: Additional Qualitative comparison between the SD3.5-M and SD3.5-M + Flow-GRPO trained with **GenEval** reward.

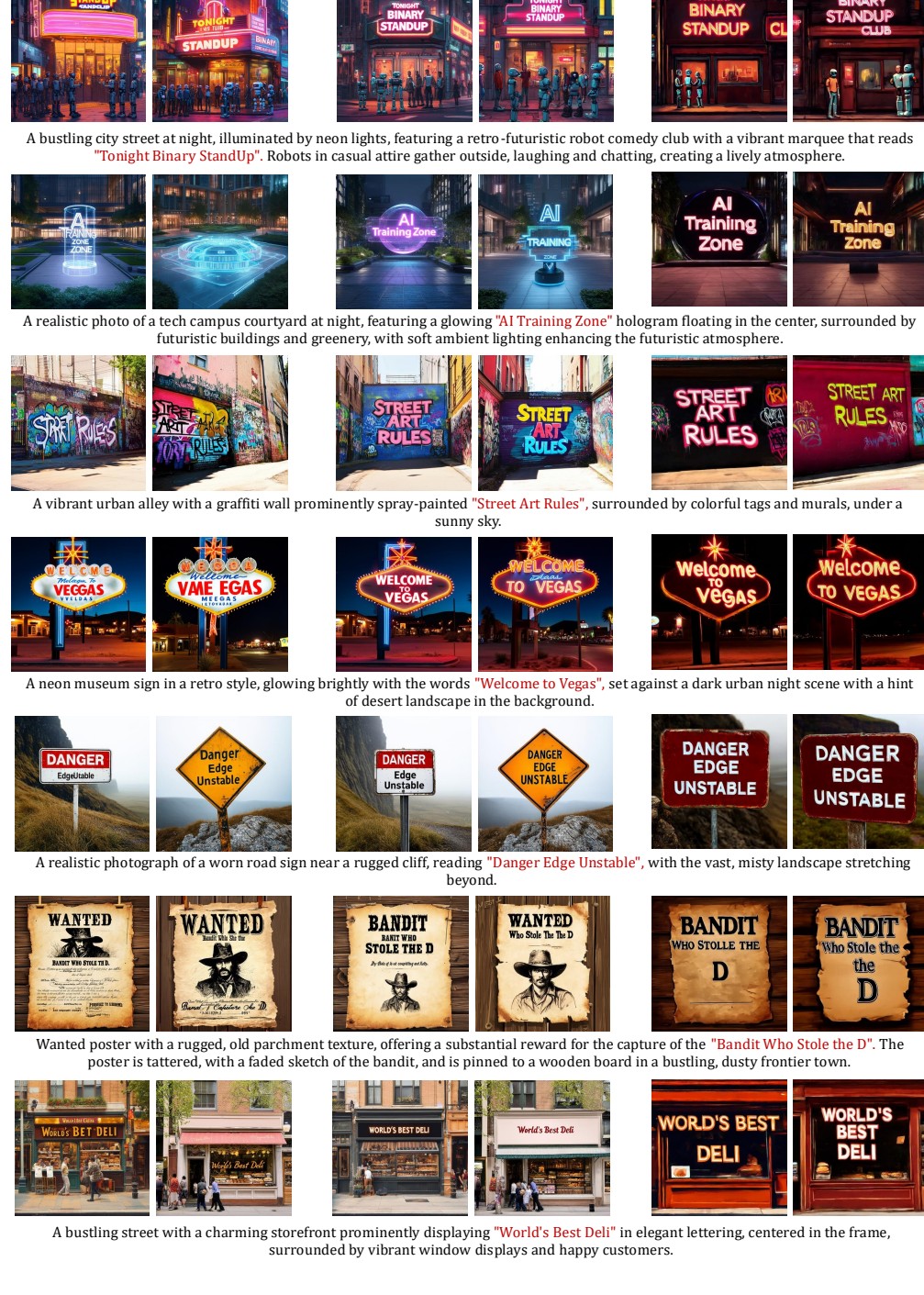

SD-3.5-M                    **Flow-GRPO**              Flow-GRPO**(w/o KL)**

A bustling city street at night, illuminated by neon lights, featuring a retro-futuristic robot comedy club with a vibrant marquee that reads "Tonight Binary StandUp". Robots in casual attire gather outside, laughing and chatting, creating a lively atmosphere.

A realistic photo of a tech campus courtyard at night, featuring a glowing "AI Training Zone" hologram floating in the center, surrounded by futuristic buildings and greenery, with soft ambient lighting enhancing the futuristic atmosphere.

A vibrant urban alley with a graffiti wall prominently spray-painted "Street Art Rules", surrounded by colorful tags and murals, under a sunny sky.

A neon museum sign in a retro style, glowing brightly with the words "Welcome to Vegas", set against a dark urban night scene with a hint of desert landscape in the background.

A realistic photograph of a worn road sign near a rugged cliff, reading "Danger Edge Unstable", with the vast, misty landscape stretching beyond.

Wanted poster with a rugged, old parchment texture, offering a substantial reward for the capture of the "Bandit Who Stole the D". The poster is tattered, with a faded sketch of the bandit, and is pinned to a wooden board in a bustling, dusty frontier town.

A bustling street with a charming storefront prominently displaying "World's Best Deli" in elegant lettering, centered in the frame, surrounded by vibrant window displays and happy customers.

Figure 14: Additional Qualitative comparison between the SD3.5-M and SD3.5-M + Flow-GRPO trained with **OCR** reward.

SD-3.5-M                    **Flow-GRPO**              Flow-GRPO(w/o KL)

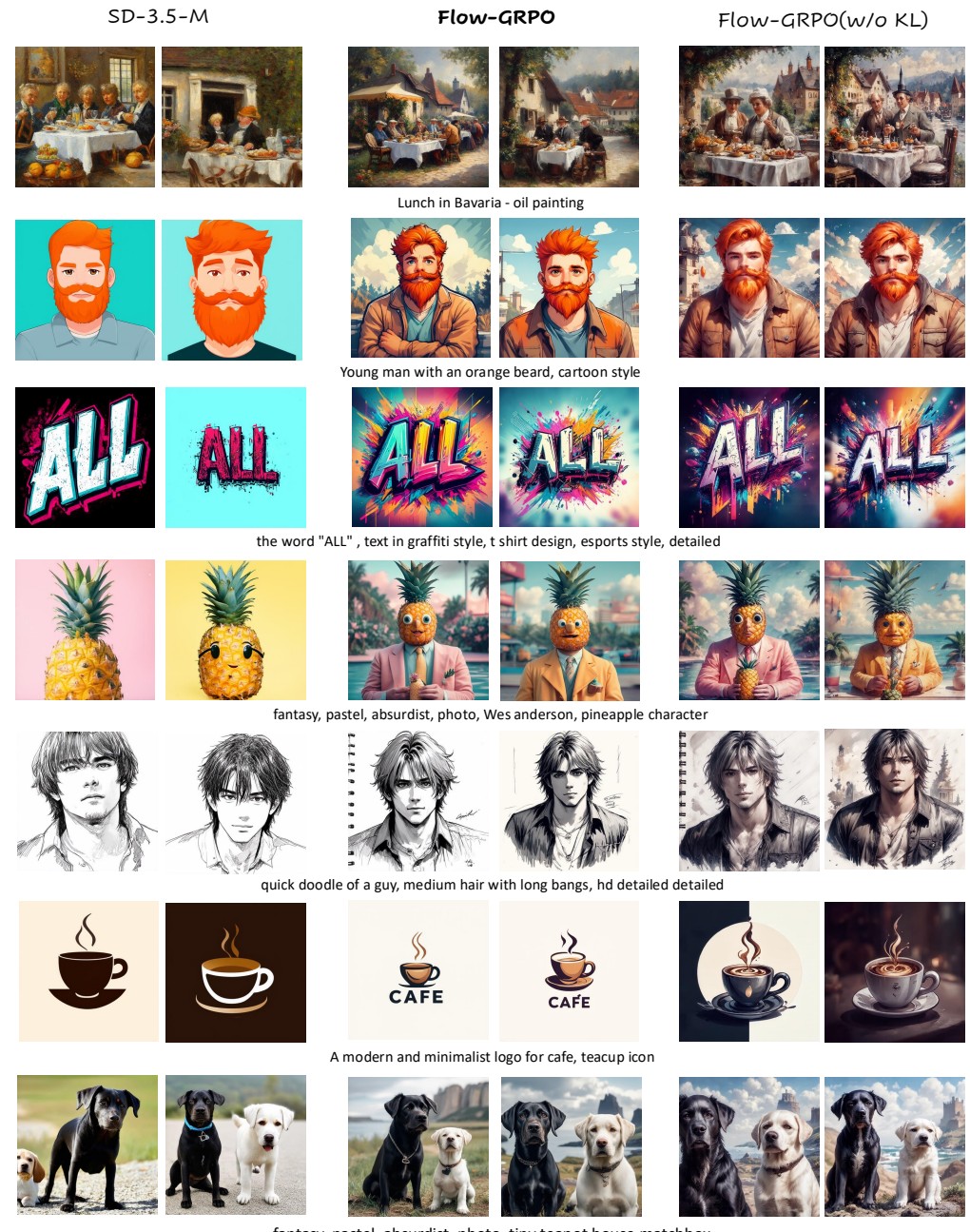

Lunch in Bavaria - oil painting

Young man with an orange beard, cartoon style

the word "ALL" , text in graffiti style, t shirt design, esports style, detailed

fantasy, pastel, absurdist, photo, Wes anderson, pineapple character

quick doodle of a guy, medium hair with long bangs, hd detailed detailed

A modern and minimalist logo for cafe, teacup icon

fantasy, pastel, absurdist, photo, tiny teapot house matchbox

Figure 15: Additional Qualitative comparison between the SD3.5-M and SD3.5-M + Flow-GRPO trained with **PickScore** reward.

**Training Process on GenEval Task** ➡

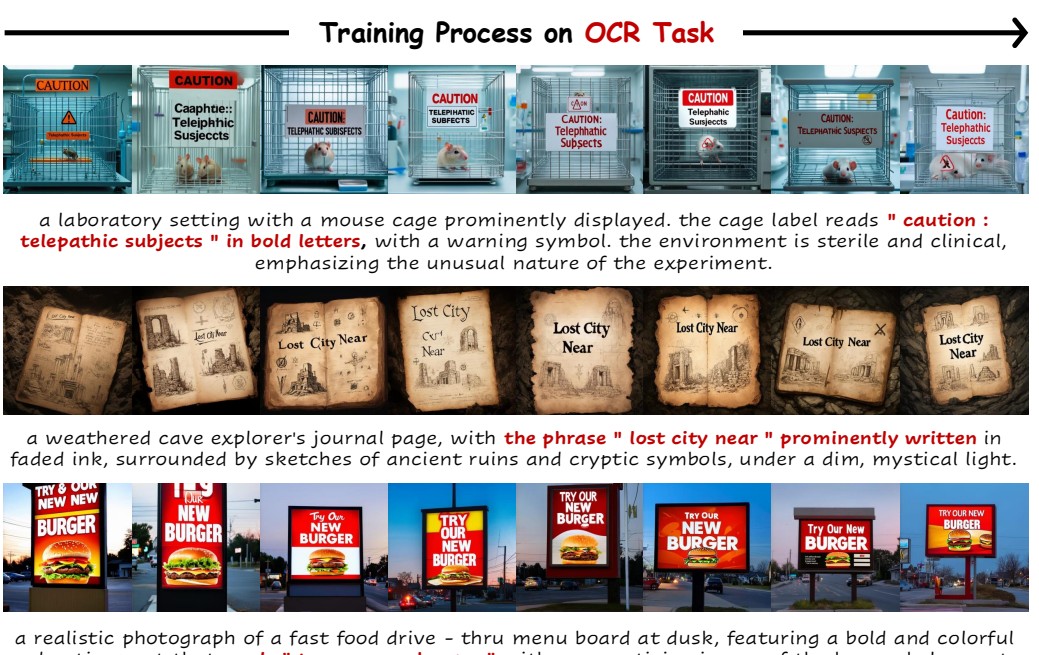

a photo of **four boats**

a photo of a cow **left of** a stop sign.

a photo of **a blue pizza** and **a yellow baseball glove**.

Figure 16: We visualize the generated samples across successive training iterations during the optimization of SD3.5-Medium on the **GenEval** task.

**Training Process on OCR Task** ➡

a laboratory setting with a mouse cage prominently displayed. the cage label reads **" caution : telepathic subjects " in bold letters,** with a warning symbol. the environment is sterile and clinical, emphasizing the unusual nature of the experiment.

a weathered cave explorer's journal page, with **the phrase " lost city near " prominently written** in faded ink, surrounded by sketches of ancient ruins and cryptic symbols, under a dim, mystical light.

a realistic photograph of a fast food drive - thru menu board at dusk, featuring a bold and colorful advertisement that **reads " try our new burger "** with an appetizing image of the burger below, set against the backdrop of a busy suburban street.

Figure 17: We visualize the generated samples across successive training iterations during the optimization of SD3.5-Medium on the **OCR** task.

**Training Process on PickScore Task** ⟶

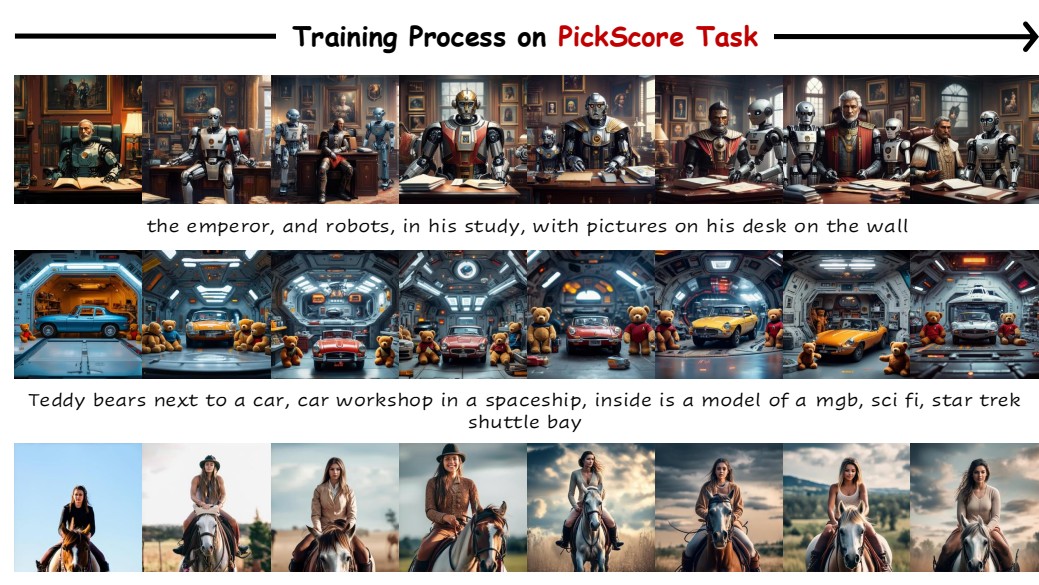

the emperor, and robots, in his study, with pictures on his desk on the wall

Teddy bears next to a car, car workshop in a spaceship, inside is a model of a mgb, sci fi, star trek shuttle bay

a woman on top of a horse

Figure 18: We visualize the generated samples across successive training iterations during the optimization of SD3.5-Medium on the **PickScore** task.

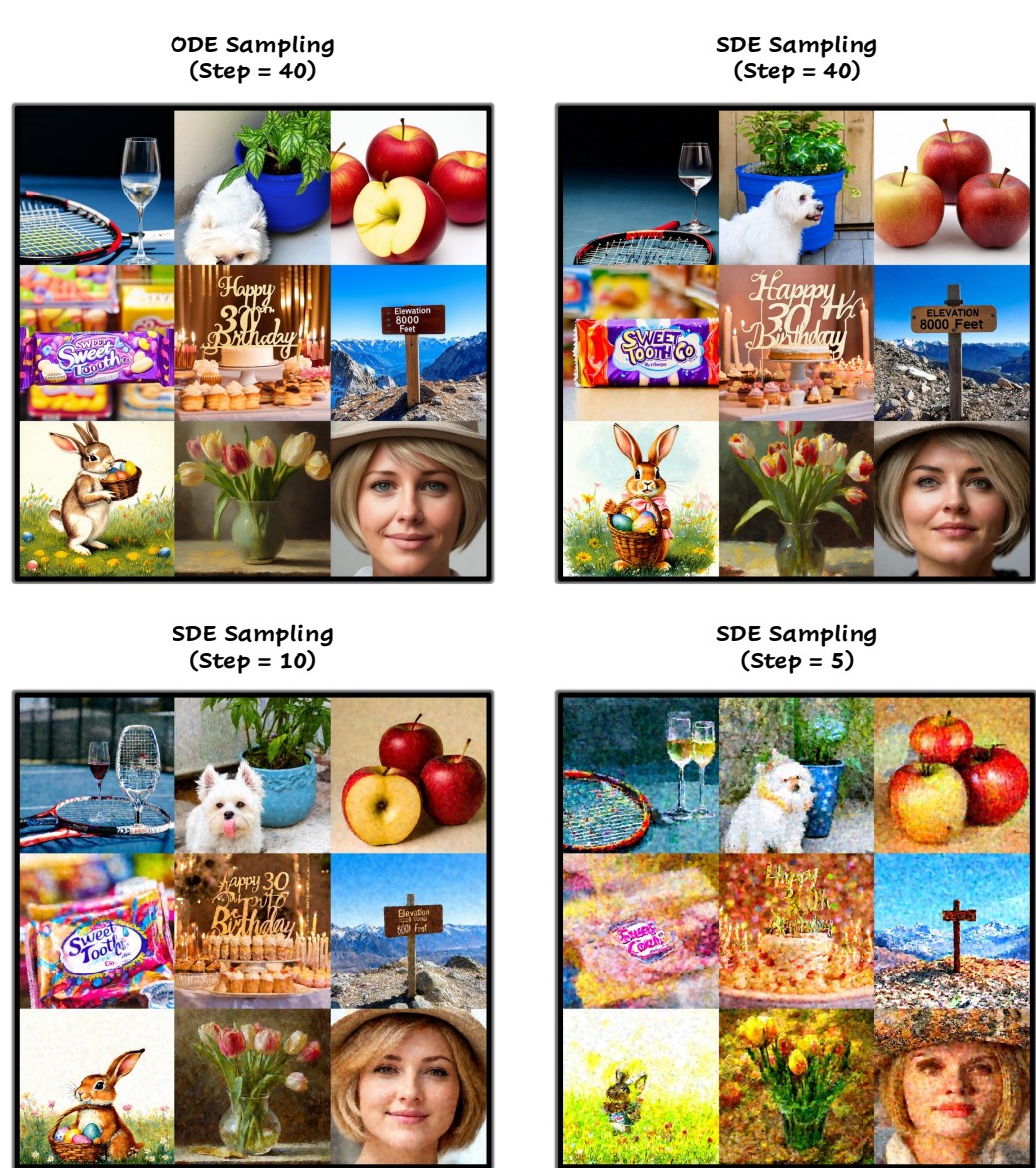

Figure 19: Visualization of training samples under difference inference settings.

