# OpenReview forum: "Flow-GRPO: Training Flow Matching Models via Online RL"
_NeurIPS.cc/2025/Conference — NeurIPS 2025 poster_

### Official Review · Reviewer_L1rz · 2025-06-26

**Clarity:** 3
**Significance:** 3
**Originality:** 3
**Rating:** 5
**Confidence:** 4

**Summary:**

This paper attempts to integrate the GRPO algorithm from LLM Online RL into Flow Matching models to enhance T2I models' capabilities in generating complex scene compositions involving multiple objects, attributes, relationships, and text rendering. To achieve this, the authors propose an ODE-to-SDE conversion that transforms the deterministic ODE into an equivalent stochastic differential equation (SDE) that matches the original model's marginal distributions across all time steps, thereby introducing the necessary stochasticity for RL exploration. Additionally, to improve RL training efficiency, the method significantly reduces denoising steps during RL training while preserving the original full step count during inference. The authors demonstrate substantial performance improvements across multiple benchmarks and claim that their approach exhibits little to no reward hacking.

**Questions:**

1. **Sampling strategy**: Why does the GRPO computation use different random noise to generate different final images for the same prompt, rather than using the same initial noise with different sampling paths? The authors should provide an additional experiment comparing different initial noise + ODE sampling paths versus the current approach.
2. **Prompt ratio justification**: In Compositional Image Generation, the authors set the prompt ratio as Position : Counting : Attribute Binding : Colors : Two Objects : Single Object = 7 : 5 : 3 : 1 : 1 : 0, but provide no justification for this specific setting nor ablation studies analyzing its impact.

**Ethical Concerns:**

["NO or VERY MINOR ethics concerns only"]

**Final Justification:**

The authors successfully addressed all my concerns, so I increase my final score to "5: Accept"

**Limitations:**

yes

**Quality:**

3

**Strengths And Weaknesses:**

## Strengths
1. The authors successfully adapt LLM Online RL algorithms to Flow Matching models, improving T2I generation capabilities for complex compositional scenes involving multiple objects, attributes, relationships, and text rendering.
2. The proposed conversion from deterministic ODE to an equivalent SDE that maintains marginal distributions across all time steps is theoretically sound. This introduces necessary stochasticity for RL exploration while preserving the model's generative characteristics.
3. The approach of significantly reducing denoising steps during RL training while maintaining full steps during inference substantially improves sampling efficiency in the RL process and accelerates training without sacrificing inference-time image quality.
4. Flow-GRPO demonstrates good generalization to unseen object categories and quantities. For instance, despite being trained on scenes with 2-4 objects, the model can generate scenes with 5-6 objects.
5. The paper is well-written with clear explanations and high-quality figures and tables.


## Weaknesses
1. The most significant limitation is the absence of comparisons with existing Diffusion/Flow Model RL methods such as Online DPO [1], SPO [2], and D3PO/DDPO [3,4]. Without these comparisons, it is difficult to assess the true advantages of GRPO over established approaches. Specifically, Online DPO could serve as a baseline by training SD3.5-M using good-bad pairs generated based on reward model scores; SPO results could provide aesthetic quality baselines; and D3PO/DDPO could serve as baselines for other reward models.
2. While Table 1 shows excellent results for Counting, Position, and Color reward scores, the paper lacks evaluation on other critical aspects such as aesthetic quality and text-image alignment. It remains unclear whether improvements in specific metrics come at the cost of degradation in others. If not, the authors should explain why GRPO better mitigates this issue compared to existing Diffusion/Flow Model RL algorithms.
3. The authors' claims about reward hacking mitigation in the abstract and introduction are misleading. Upon careful reading, the alleviation of reward hacking stems primarily from reward ensembling and KL constraints rather than the GRPO algorithm itself. This raises questions about the paper's core contributions.
4. The evaluation is restricted to SD3.5-M, leaving the effectiveness on other Flow-based models (e.g., HunyuanVideo, Flux) unclear. Additionally, the exclusive focus on Flow-based models is questionable, as Diffusion-based models can also benefit from Online RL approaches.
5. Despite Table 4 demonstrating some generalization in categories and quantities, this capability appears constrained. For example, Table 3 shows that Flow-GRPO performs worse than the original SD3.5-M on texture generation.
6. The technical contributions are relatively incremental. ODE-to-SDE conversion has been explored in prior works [5,6], and the authors make no algorithmic innovations to GRPO itself. While demonstrating GRPO's applicability to Flow Models, the absence of Online RL baselines and failure to establish GRPO's superiority over other Online RL algorithms in image generation (noting that reward hacking mitigation is not GRPO's achievement) weakens the contribution.

## Summary
While the paper presents an interesting application of GRPO to Flow Matching models, the significant limitations in experimental evaluation, particularly the absence of critical baselines and overclaimed contributions regarding reward hacking, substantially weaken its impact. The work would benefit from comprehensive comparisons with existing methods and more honest characterization of its contributions.

### **Reference**
- [1] Diffusion Model Alignment Using Direct Preference Optimization, CVPR 2024
- [2] Aesthetic Post-Training Diffusion Models from Generic Preferences with Step-by-step Preference Optimization, CVPR 2025
- [3] Direct Preference for Denoising Diffusion Policy Optimization, CVPR 2024
- [4] Training Diffusion Models with Reinforcement Learning, ICLR 2024
- [5] Semantic Image Inversion and Editing using Stochastic Rectified Differential Equations, ICLR 2025
- [6] Stochastic Interpolants: A Unifying Framework for Flows and Diffusions, arXiv'23

---

> ### Author Rebuttal · Authors · 2025-07-31
>
> We sincerely appreciate your insightful comments and efforts in reviewing our manuscript. We respond to each of your comments one-by-one in what follows. Due to space limits, part of the responses to `[W1` is included in other reviewers' rebuttals. We apologize for the inconvenience and kindly refer the reviewer to those sections for details.
>
>
> ---
>
> ` [W1] Comparison with other RL methods`
>
>
> Thanks for your suggestions. First, in Appendix C.2, we already provide the comparison with other RL baselines, including offline SFT/DPO/RWR and their online variants. Results show that Flow-GRPO trains more stably than other RL baselines and ultimately attains the highest evaluation reward. In addition, SPO is a very insightful work, but it requires a step-aware preference model to label win-lose pairs. In contrast, our method only needs a reward signal from the final, clean image. The final-image equivalent of SPO is DPO, which we have already compared against. We will cite SPO in our revision and clarify this distinction.
>
>
> Beyond the existing results, we have also conducted additional comparative experiments, these results will be included in the revision. Due to space limitations, we kindly refer the reviewer to `Reviewer JifQ's ［W3］and [Q1 & Q2]` for further details. We will include this information in the revised version of the main paper. We summarised below:
>
>
> ### 1. Comparison with other RL baselines on GenEval.
> Please see `Reviewer JifQ's ［W3］`
>
> ### 2. Comparison with DDPO on PickScore.
> Please see `Reviewer JifQ's ［W3］`
>
> ### 3. Comparison with ORW on CLIP Score.
> Please see `Reviewer JifQ's [Q1 & Q2]`
>
> ---
>
> ` [W2] Whether improvements in specific metrics come at the cost of degradation in others `
>
> In Table 2, we have already covered the other critical aspects you mentioned. Specifically, it reports image quality metrics such as Aesthetic and DeQA, along with human preference metrics like ImageReward, PickScore, and UnifiedReward. **Flow-GRPO improves the task-specific metrics without degrading these additioanl metrics.**
>
>
> ---
>
> ` [W3] The alleviation of reward hacking stems primarily from GRPO itself.`
>
> We emphasize that we do not claim to "invent" the KL constraint, instead our results show that a carefully tuned KL penalty can match the high rewards of a KL-free run while preserving image quality, albeit at the cost of slightly longer training. Importantly, the KL term is not viewed as a necessary component of GRPO, previous studies [1] have even reported improved scores after removing it. **Our contribution is that, through comprehensive ablation studies, we demonstrate that the KL constraint is critical to the Flow‑GRPO algorithm, enabling the reward to keep rising without any collapse in fidelity or diversity.** This is not a general consensus in online RL for generative models, we believe our finding offers useful empirical guidance to the community.
>
>
> [1]. DAPO: An Open-Source LLM Reinforcement Learning System at Scale.
>
> ---
>
>
> ` [W4] The effectiveness on other Flow-based models/diffusion-based models`
>
>
> Thank you for this valuable suggestion. We have run Flow-GRPO on FLUX.1-Dev using PickScore as the reward signal. The reward curve rises steadily throughout the training without noticeable reward hacking. We report:
>
> **(1) The reward value over the training process**
>
> $$
> \begin{array}{lccccc}
> \hline
> \text{Method} & \text{Step 0} & \text{Step 240} & \text{Step 480} & \text{Step 720} & \text{Step 960} \newline
> \hline
> \text{FLUX.1-Dev + Flow-GRPO}  & 21.94 & 22.78 & 22.98 & 23.19 & 23.36   \newline
> \hline
> \end{array}
> $$
>
> **(2) Comparison between *FLUX.1-Dev* and *FLUX.1-Dev + Flow-GRPO* on DrawBench.**
>
> $$
> \begin{array}{lccccc}
> \hline
> \text{Model} & \text{Aesthetic} & \text{DeQA} & \text{ImageReward} & \text{PickScore} & \text{UnifiedReward} \newline
> \hline
> \text{FLUX.1-Dev} & 5.71 & 4.31 & 0.85 & 22.62 & 3.65   \newline
> \text{FLUX.1-Dev + Flow-GRPO}  & 6.02 & 4.24 & 1.32 & 23.97 & 3.81    \newline
> \hline
> \end{array}
> $$
>
>
> Although our experiment mainly focuses on flow-matching models, Flow-GRPO could be applied directly in diffusion models without modification. If a diffusion model were used instead, this stochasticity is already built in their DDPM-style sampling, so the same pipeline in Flow-GRPO can be directly utilized in diffusion models' optimization.
>
>
> ---
>
> ` [W5] Flow-GRPO performs worse than the original SD3.5-M on texture generation.`
>
> In T2I‑CompBench++, the “texture generation” evaluates the ability to render material appearance (e.g., plastic, metallic, and wooden). **This axis is essentially orthogonal to the objectives we optimise in GenEval**, which focus on counting, color binding, and 2D spatial relations, and our training prompts contain no vocabulary or descriptions of object material. Given that the model receives no reward signal in that direction, a slight decline from 0.7338 to 0.7236 is unsurprising and negligible when set against the broader gains Flow‑GRPO achieves on its target tasks.
>
>
> ---
>
>
> ` [W6] The technical contributions are relatively incremental `
>
>
> Thank you for the comment. Our paper does not aim to propose a novel RL algorithm but instead aims to address a critical open question: *Can contemporary generative models be improved through online RL, and how to achieve this for the flow-matching backbones now dominating SOTA generation models.* Before our work, PPO/GRPO have shown remarkable success in understanding tasks with LLMs, while their applicability to flow-matching generation models has not yet been demonstrated. We show that with carefully chosen modifications, flow-matching models can profit from online RL, yielding measurable gains in multi generation tasks. Our results provide key design guidelines and empirical evidence for applying online RL to flow-based generators, opening new directions for future research in this area. We summarise our key contributions as follows:
>
> - **We establish a well-defined MDP for flow-matching generators.** GRPO presupposes stochastic, step-by-step transitions so that a policy $p_{\theta}(a_t \mid s_t)$ is well defined, yet a standard FM model evolves deterministically under its ODE formulation. By leveraging the connections between FM models and score-based generative models, we convert the ODE into an SDE with controlled noise supplying the required randomness. This injects exactly the stochasticity needed to instantiate the MDP without altering the model's marginal distribution.
> - **We introduce several practical strategies that make online RL both efficient and robust.** We find that online RL does not require the standard long timesteps for training sample collection. By using fewer denoising steps during training and retaining the original steps during testing, we can significantly accelerate the training process with 4x speedup. Additionally, we show that the KL constraint could suppress reward hacking, reaching the high reward while preserving image quality and diversity.
> - **We provide empirical validation of the effectiveness of Flow-GRPO.** We evaluate Flow-GRPO on T2I tasks with various reward types. This includes: 1) verifiable rewards for compositional generation and visual text rendering; 2) model-based rewards for human-preference alignment. Across all settings, Flow-GRPO yields consistent improvements, demonstrating that the proposed framework is a promising path for enhancing SOTA generative models.
>
>
>
> ---
>
> ` [Q1] Why use different random noise? The authors should provide an additional experiment comparing different initial noise + ODE sampling paths versus the current approach `
>
> We initialize each rollout with different random noise to increase exploratory diversity during RL training. We perform an additioanl ablation to confirm this claim. With SD-3.5-M as the base model and PickScore as the reward, we compare Flow-GRPO with different initial noise against Flow-GRPO with the same initial noise. The variant with different noise consistently achieved high rewards during the training process. The detailed results are reported below:
>
>
> $$
> \begin{array}{lccccc}
> \hline
> \text{Method} & \text{Step 0} & \text{Step 240} & \text{Step 480} & \text{Step 720} & \text{Step 960} \newline
> \hline
> \text{Flow-GRPO, same inital noise}  & 21.72 & 22.53 & 22.98 & 23.12 & 23.30   \newline
> \textbf{Flow-GRPO, different inital noise}  & \textbf{21.72} & \textbf{22.53} & \textbf{23.01} & \textbf{23.31} & \textbf{23.41}   \newline
> \hline
> \end{array}
> $$
>
>
> A comparison based on “different noise + pure ODE sampling” is inapplicable. GRPO requires a stochastic, step‑by‑step transition model so that the policy $p_{\theta}(a_t \mid s_t)$ is well defined. ODE sampling is deterministic, therefore it does not furnish an MDP; under that setting the GRPO objective cannot be optimised.
>
> ---
>
>
> `  [Q2] Prompt ratio justification `
>
> The prompt ratio was selected empirically according to the relative difficulty of each task. The base SD-3.5-M model handled single-object and two-object scenes almost flawlessly, yet struggled with position, counting, and attribute binding. We therefore allocated a larger portion of training prompts to the harder categories and a smaller portion to the easier ones. This trick does not affect our contributions and conclusions, as each RL baseline is trained with the identical backbone and the same training prompts.

---

> > ### Comment · Reviewer_L1rz · 2025-08-05
> >
> > Thank you for the authors' efforts in preparing the rebuttal. I have carefully reviewed your responses but still have the following concerns:
> >
> > ### [W1] Comparison with other RL methods
> > 1. The authors explicitly mention in Appendix C.1 (lines 343-344) that "online variants update their data collection model every 40 steps." However, I could not find information regarding the update frequency for Flow-GRPO's data collection in either the main paper or supplementary materials. If Flow-GRPO updates data at every step while other baseline methods update every 40 steps, this comparison would be unfair. The authors should clearly specify this in the paper.
> > 2. The authors provided comparisons with relatively weak baseline methods such as Online DPO and RWR, but lack comparisons with other online RL algorithms like ImageReward/SPO. Based on the current results, I still cannot determine whether GRPO offers substantial improvements over existing Diffusion RL algorithms.
> > 3. I hope the authors can refine the Table 2 in future versions by clearly indicating which metrics are used as reward signals during training versus those used solely for evaluation during inference. If these metrics are used in both training and inference phases, the results may not be credible due to reward hacking.
> >
> > ### [W2] Whether improvements in specific metrics come at the cost of degradation in others
> > The authors have misunderstood my concern. Flow-GRPO's reward is defined using the metrics employed in the paper: (1) Compositional Image Generation, (2) Text Rendering, (3) Human Preference Alignment, and (4) Image Quality Evaluation. The authors trained and evaluated using the same reward metrics, which risks metric hacking. A reasonable validation would be to use Metrics 1-3 as the reward function for training and observe whether there is performance degradation on Metric 4, thereby verifying whether reward hacking has truly been mitigated. Alternatively, the authors should demonstrate under this setting that GRPO outperforms existing online RL algorithms, rather than using the same specific reward function for both training and testing as currently done.
> >
> > ### [W3] The alleviation of reward hacking stems primarily from GRPO itself
> > The authors claim their method can mitigate reward hacking but do not clarify that this is due to KL regularization and reward ensemble rather than the GRPO algorithm itself. This may mislead readers into thinking GRPO inherently solves reward hacking, which is not the case. The authors need to clearly specify this in the abstract and introduction (i.e., GRPO CANNOT alleviate reward hacking, but KL and Reward Ensemble can). Furthermore, this raises an important question: what exactly are the advantages of GRPO compared to existing online RL algorithms?
> >
> > ### [W4, W5, W6]
> > I am satisfied with the authors' responses and have no further questions on these points.
> >
> > ### Summary
> > 1. Although the authors' rebuttal has addressed some concerns to a certain extent, my most important concerns about this paper remain insufficiently resolved:
> > 2. The online RL baselines for comparison are still inadequate (missing baselines like ReFL/SPO)
> > Using the same metrics for both training and validation phases fails to demonstrate improvement in some metrics without compromising others
> > 3. The advantages of GRPO over existing online RL algorithms remain unclear, and the reward hacking mitigation is not inherently due to GRPO itself
> > 4. The overclaim on alleviation reward hacking
> >
> > Therefore, I prefer to maintain the current score and may revise it based on subsequent discussions.

---

> > > ### Author Response · Authors · 2025-08-05
> > > **Response to Reviewer L1rz (Part 1/2)**
> > >
> > > Thank you again for the thoughtful comments. We are glad our rebuttal has addressed some of your concerns; for the remaining issues, we respond point-by-point below.
> > >
> > > > [W1-1] The authors explicitly mention in Appendix C.1 (lines 343-344) that "online variants update their data collection model every 40 steps." However, I could not find information regarding the update frequency for Flow-GRPO's data collection in either the main paper or supplementary materials. If Flow-GRPO updates data at every step while other baseline methods update every 40 steps, this comparison would be unfair. The authors should clearly specify this in the paper
> > >
> > >
> > > Thank you for the comment. Flow-GRPO updates its data collection at every step. **We also attempted step-wise updates for the DPO/RWR baselines, but training became unstable and quickly collapsed**, metrics such as GenEval dropped to near zero and PickScore to ~17 because the models cannot produce valid images. Therefore we switched to periodic updates as reported in Appendix C.1, which yielded better results. This choice is consistent with prior observations for DDPO[1], their Fig. 11 states that "more frequent interleaving is beneficial up to a point, **after which it causes performance degradation**".
> > >
> > > [1] Training Diffusion Models with Reinforcement Learning, ICLR 2024.
> > >
> > > ---
> > >
> > > > [W1-2] The authors provided comparisons with relatively weak baseline methods such as Online DPO and RWR, but lack comparisons with other online RL algorithms like ImageReward/SPO. Based on the current results, I still cannot determine whether GRPO offers substantial improvements over existing Diffusion RL algorithms.
> > >
> > > Thank you for the comment. As we noted earlier, SPO[2] is a DPO variant that requires a step-aware preference model, while **our adopted reward models are step-agnostic and thus do not satisfy this requirement**. ImageReward[3] assumes a differentiable reward, whereas our targets like GenEval and OCR accuracy are non-differentiable, **making that baseline inapplicable in our setting**. Both works are very insightful, but their special requirements on the reward model prevent a direct comparison with Flow-GRPO or with DPO/RWR baselines under our current evaluation protocol. We will expand the discussion of these methods in the revision.
> > >
> > > Aside from these RL algorithms that impose special requirements on the reward model, we already compare against a broad suite of RL baselines, including SFT/DPO[4, 5]/RWR[5](offline and online), DDPO[1], and ORW[6], which we believe provides a sufficiently representative comparison with recent diffusion/flow-based RL methods.
> > >
> > > [2] Aesthetic Post-Training Diffusion Models from Generic Preferences with Step-by-step Preference Optimization, CVPR 2025.
> > > [3] ImageReward: Learning and Evaluating Human Preferences for Text-to-Image Generation, NeurIPS 2023.
> > > [4] Diffusion Model Alignment Using Direct Preference Optimization, CVPR 2024.
> > > [5] Improving Video Generation with Human Feedback, arXiv 2025.
> > > [6] Online Reward-Weighted Fine-Tuning of Flow Matching with Wasserstein Regularization. ICLR 2025.
> > >
> > > ---
> > >
> > > > [W1-3 & W2] Whether improvements in specific metrics come at the cost of degradation in others
> > >
> > > **We believe there is a misunderstanding of Table 2.** In our setup, the **training rewards** are exactly Metrics 1–3, including GenEval, OCR accuracy, and PickScore, which are marked as “Task Metrics” in the table. The other metrics (e.g., Aesthetic, DEQA, ImageReward, and the others) are **evaluation-only**, **and most are not used for training.**
> > >
> > > Our validation actually follows your proposed protocol: we train with Metrics 1–3 and then check for degradation on the held-out quality and preference metrics. As Table 2 shows, there is no such degradation; in many cases these held-out metrics improve.
> > >
> > > ---

---

> > > ### Author Response · Authors · 2025-08-05
> > > **Response to Reviewer L1rz (Part 2/2)**
> > >
> > > ---
> > >
> > >
> > > > [W3] The authors claim their method can mitigate reward hacking but do not clarify that this is due to KL regularization and reward ensemble rather than the GRPO algorithm itself. This may mislead readers into thinking GRPO inherently solves reward hacking, which is not the case. The authors need to clearly specify this in the abstract and introduction (i.e., GRPO CANNOT alleviate reward hacking, but KL and Reward Ensemble can). Furthermore, this raises an important question: what exactly are the advantages of GRPO compared to existing online RL algorithms?
> > >
> > > Thank you for the kindly suggestion, **we will clearly specify in the abstract and introduction that GRPO does not itself mitigate reward hacking.** In our experiments, KL regularization are the mechanisms that prevent reward hacking.
> > >
> > > ---
> > >
> > > Regarding `what exactly are the advantages of GRPO compared to existing online RL algorithms`. We answer it in two parts.
> > >
> > >
> > > ### 1. How GRPO differs from existing baselines
> > >
> > > Methods like DPO and its variants SPO are RL-free preference-optimization approaches (as they emphasize themself as "rl-frree algorithms" in the original DPO paper[7]). And DDPO, by contrast, is an RL method designed for diffusion bascknones. To compare on a flow-matchng model, we also adapted DDPO via our ODE-to-SDE conversion and evaluated with SD-3.5-M and PickScore:
> > >
> > >
> > > $$
> > > \begin{array}{l|ccccc}
> > > \hline
> > > \text{Method} & \text{Step 0} & \text{Step 240} & \text{Step 480} & \text{Step 720} & \text{Step 960} \newline
> > > \hline
> > > \text{DDPO (with ODE-to-SDE)}  & 21.72 & 21.91 & 22.01 & 21.93 & 17.64   \newline
> > > \textbf{Flow-GRPO} & \textbf{21.72} & \textbf{22.53} & \textbf{23.01} & \textbf{23.31} & \textbf{23.41}   \newline
> > > \hline
> > > \end{array}
> > > $$
> > > We observe that DDPO's reward rises more slowly than Flow-GRPO's and collapses in the late stages, whereas Flow-GRPO trains steadily and continues to improve throughout the training process.
> > >
> > >
> > > [7] Direct Preference Optimization: Your Language Model is Secretly a Reward Model, NeurIPS 2023.
> > >
> > > ### 2. Why introduce GRPO for flow-matching models
> > >
> > > **Compared to DPO-like methods.** In the field of large language models(LLMs), classic policy-gradient algorithms (GRPO/PPO) have repeatedly outperformed DPO-style algorithms both in reasoning [8], human preference alignment [9], where “PPO is able to surpass other alignment methods in all cases”, and general domains [10], which states that “PPO outperforms DPO in general domains.” In our flow-matching experiments, Flow-GRPO likewise surpasses DPO, RWR, SFR, and their online variants in both stability and final reward.
> > >
> > > **Compared to PPO.** GRPO is simpler and cheaper to run. GRPO does not require an explicit value function to estimate advantages, instead it computes advantages via groupwise comparisons. Moreover, DDPO can be viewed as a PPO-like method without a value network. Our results on DDPO show that removing the value model induces instability. GRPO avoids that issue by computing advantages from groups. And the standard PPO with a value model introduces many extra hyperparameters and is widely known to be difficult to train robustly in practice, even in LLM [11], which notes that “acquiring a reliable value model is inherently challenging for PPO.” Reflecting this, the latest Qwen3 [11] model adopts GRPO over PPO.
> > >
> > > [8]. DeepSeek-R1: Incentivizing Reasoning Capability in LLMs via Reinforcement Learning, arXiv 2025.
> > > [9]. Is DPO Superior to PPO for LLM Alignment? A Comprehensive Study, arxiv 2024.
> > > [10]. Unpacking DPO and PPO: Disentangling Best Practices for Learning from Preference Feedback, arxiv 2024.
> > > [11]. Group Sequence Policy Optimization, arXiv 2025.
> > >
> > >
> > > ---
> > >
> > > Taken together, we believe that GRPO offer clear advantages over DPO/RWR/PPO baselines, our experiments also support this observation. Accordingly, introducing GRPO into the post-training of flow-matching models is a concrete and practically meaningful contribution.

---

> ### Comment · Reviewer_L1rz · 2025-08-05
>
> Thank you for the authors' efforts in addressing the concerns raised in my initial review.
>
> ### [W1]
> The authors should conduct fair comparisons with ReFL using differentiable reward functions (e.g., HPSv2 or aesthetic models), otherwise it is difficult to demonstrate that GRPO outperforms existing ReFL algorithms in terms of performance.
>
> ### [W2, W3]
> I appreciate the authors' clarification and efforts made in this regard. This discussion has effectively addressed my concerns. However, I still recommend that the authors clearly indicate in Table 2 which metrics are used for training versus evaluation, and update future versions with the details mentioned in W1 (data update frequencies of different methods) as well as a clearer description of reward hacking.
>
> ### [W4]
> The authors' response in this section only demonstrates the advantages of "GRPO" over "PPO/DPO" but fails to explain why "GRPO" is more suitable for "Diffusion models" compared to existing reinforcement learning algorithms (such as ReFL). Furthermore, while ReFL requires differentiable reward functions, it does not need multiple rollouts like GRPO, making it difficult to intuitively understand where GRPO's advantages lie.
>
>
> The authors have resolved 2 of my main concerns in this response, therefore I am raising my score to Borderline Reject for now. However, I still have questions regarding W1 and W4, and hope the authors can actively respond to these remaining concerns.

---

> > ### Author Response · Authors · 2025-08-06
> > **Response to Reviewer L1rz (Part 1/2)**
> >
> > Thank you again for your continued engagement and thoughtful feedback. We’re glad that some of your concerns have been resolved, and we now address the new issues you’ve raised below.
> >
> > ---
> >
> > > [W1 & W4] GRPO vs. ReFL
> >
> >
> > We agree that a direct comparison with ReFL is crucial for demonstrating the advantages of GRPO. Our response aims to clarify three key points:
> > 1.  When a differentiable reward is available, **GRPO performs on par with or even better than ReFL**, demonstrating its robustness and generality.
> > 2.  The primary advantage of GRPO lies in its ability to leverage **non-differentiable reward functions**, which are often more powerful, general-purpose, and easier to develop than their differentiable counterparts.
> > 3.  There are significant practical and computational barriers to applying ReFL, especially for modern, high-performance video diffusion models.
> >
> >
> > ### 1. Direct Comparison with ReFL
> >
> > To directly address the reviewer's request for a fair comparison, we implemented ReFL using a differentiable reward function (PickScore [1]) and compared its performance against GRPO using the same reward. We track the evaluation reward over the entire training process, as shown below:
> >
> > $$
> > \begin{array}{l|ccccc}
> > \hline
> > \textbf{Method} & \textbf{Step 0} & \textbf{Step 240} & \textbf{Step 480} & \textbf{Step 720} & \textbf{Step 960} \newline
> > \hline
> > \text{ReFL} & 21.70 & 22.10 & 22.46 & 22.69 & 22.81 \newline
> > \textbf{Flow-GRPO} & 21.72 & 22.53 & 23.01 & 23.31 & 23.41 \newline
> > \hline
> > \end{array}
> > $$
> >
> > **Note:** *Since our paper does not use HPSv2 or aesthetic models in the human preference task, we chose PickScore, a similar reward model already used in our experiments, to reduce the workload. If the reviewer prefers, we are happy to compare ReFL and Flow GRPO using HPSv2 or aesthetic models as well.*
> >
> >
> > Following ImageReward [2], we back-propagate gradients to a randomly-picked latter timestep $t \in [30, 40]$ in the denoising process. The results show that GRPO outperforms ReFL when a differentiable reward is available. We acknowledge that directly using ImageReward’s recommended ReFL hyperparameters may not yield the best performance on the SD3.5 model, and we were unable to conduct extensive hyperparameter tuning within the limited rebuttal period. Nevertheless, this experiment at least demonstrates that GRPO does not compromise performance in domains where ReFL is applicable. More importantly, as discussed below, GRPO’s true strength lies in its ability to operate in scenarios where differentiable rewards are unavailable, impractical, or suboptimal.
> >
> >
> > ### 2. The Core Advantage: Compatibility with Advanced Non-Differentiable Rewards
> >
> > The most significant advantage of GRPO is that it does not require the reward function to be differentiable. This unlocks the ability to use state-of-the-art Vision-Language Models (VLMs) directly as reward providers. This approach has several profound benefits that are inaccessible to ReFL:
> >
> > * **Sophisticated, General-Purpose Rewards:** A VLM can execute a complex, human-like evaluation process. For instance, given a prompt, a VLM can first deconstruct it into key criteria, engage in a chain-of-thought process to verify each criterion in the generated image, and then provide a holistic score. This creates a highly versatile reward that can serve diverse tasks, from text-to-image generation to complex instruction-based image editing, using a single, unified model.
> > * **Future-Proof and Zero-Cost Updates:** The field of VLMs is advancing at a breathtaking pace. By using a VLM as the reward source, our framework automatically benefits from these improvements. As VLMs become more capable, our reward model becomes stronger without any additional training data or computational cost.
> >
> > In contrast, differentiable reward models, as required by ReFL, face significant limitations:
> >
> > * **Prohibitive Training and Data Costs:** Creating a high-quality, differentiable reward model requires training on massive, manually annotated human preference datasets. This is incredibly expensive and time-consuming. This cost is reflected in the fact that widely-used open-source rewards like HPSv2 and PickScore were trained on images from older models (e.g., Stable Diffusion 2.0). They are already lagging behind the quality of modern generators like FLUX, severely limiting the effectiveness of post-training.
> > * **The VAE Memory Bottleneck:** Differentiable rewards typically operate on the pixel level. To fine-tune a diffusion model, which operates in the latent space, gradients must be backpropagated through the VAE decoder. This process is **extremely memory-intensive**. This bottleneck is even more severe for video generation. One potential solution is to define rewards directly in the latent space. However, since latent spaces are often model-specific, designing a universal, model-agnostic latent reward model remains infeasible.

---

> > ### Author Response · Authors · 2025-08-06
> > **Response to Reviewer L1rz (Part 2/2)**
> >
> > ### 3. A Practical Example: Image Editing
> >
> > To demonstrate GRPO's practical advantage, we tackled the task of improving image editing capabilities. The community currently lacks any established differentiable reward model for this specific task; existing models (ImageReward, PickScore, HPSv2) are all designed for text-to-image alignment.
> >
> > With GRPO, we can bypass this limitation entirely by using GPT-4.1 directly as a reward function for editing quality. This is a scenario where ReFL is simply not applicable. As shown below, applying GRPO (**Flow-GRPO**) to our base editing model (**SD3.5-Kontext**) yields significant improvements across multiple standard editing benchmarks.
> >
> > $$
> > \begin{array}{l|ccc}
> > \hline
> > \textbf{Method} & \textbf{GenEval} & \textbf{GEdit [3]} & \textbf{ImgEdit [4]} \newline
> > \hline
> > \text{SD3.5-Kontext} & 0.868 & 5.624 & 3.36 \newline
> > \textbf{SD3.5-Kontext + Flow-GRPO} & \textbf{0.931} & \textbf{5.913} & \textbf{3.46} \newline
> > \hline
> > \end{array}
> > $$
> >
> > **Note:** *SD3.5-Kontext is our internal editing model based on SD3.5. It is smaller but achieves similar performance to the public FLUX-Kontext model.*
> >
> >
> >
> > In addition, a major trend in the community is the development of unified models that integrate both generative and understanding capabilities (e.g., recent models like Bagel [5]). ReFL is not suitable for fine-tuning such architectures, as its gradient-based approach cannot update the language model components. GRPO, however, is model-agnostic and can provide a unified reinforcement learning signal to jointly optimize both the LLM and the diffusion model. This makes GRPO a viable and powerful method for the end-to-end alignment of these next-generation unified models.
> >
> > In summary, GRPO is a more general and powerful framework than ReFL for aligning diffusion models. It avoids the severe data and computational bottlenecks of differentiable rewards while unlocking the vast potential of modern, non-differentiable VLMs, leading to superior performance in practical, real-world applications like image editing.
> >
> > [1]. Pick-a-Pic: An Open Dataset of User Preferences for Text-to-Image Generation, NeurIPS 2023.
> >
> > [2]. ImageReward: Learning and Evaluating Human Preferences for Text-to-Image Generation, NeurIPS 2023.
> >
> > [3]. Step1X-Edit: A Practical Framework for General Image Editing, arXiv 2025.
> >
> > [4]. ImgEdit: A Unified Image Editing Dataset and Benchmark, arXiv 2025.
> >
> > [5]. Emerging Properties in Unified Multimodal Pretraining, arXiv 2025.

---

> > > ### Comment · Reviewer_L1rz · 2025-08-06
> > >
> > > Thanks to the reply and the experimental results, my concerns have been completely resolved, so I will raise my score to "5: Accept", and hope that the author can update the discussion and experimental results of the Rebuttal stage into the final version.

---

> > > > ### Author Response · Authors · 2025-08-07
> > > >
> > > > Thank you so much for your quick response and for helping improve our work. We truly appreciate your insightful feedback, which is invaluable in improving the quality of our paper. We will be sure to incorporate the discussion and experimental results from the rebuttal stage into the final version of the paper.

---

### Official Review · Reviewer_JifQ · 2025-06-30

**Clarity:** 1
**Significance:** 2
**Originality:** 3
**Rating:** 3
**Confidence:** 4

**Summary:**

This paper proposed Flow-GPRO, which adopts the original GPRO technique for pre-trained flow-based models. Specifically, the authors transformed the flow ODE into the corresponding SDE and then applied the standard GPRO technique. Experimental results on text-to-image alignment demonstrated the superior performance compared to the original model.

**Questions:**

See the weaknesses listed above. In addition:
- The CLIP score is the standard evaluation metric for text-to-image alignment. Can you provide additional results for the CLIP score?
- Can you compare the proposed method with more recent online RL approaches like ORW mentioned in the Weakness section?

**Ethical Concerns:**

["NO or VERY MINOR ethics concerns only"]

**Final Justification:**

As I have mentioned in my review and additional response to the authors' rebuttal, the original paper **overclaimed** itself as the first online RL approach for flow models, but surprisingly **did not provide any comparison** with existing online RL baselines. The new results in the rebuttal make more sense, but require a **significant rewrite** of the current paper, as **none of the original baseline results are meaningful**.

Based on my understanding of the NeurIPS guideline, the paper should not be significantly different from the initial submission. In view of the above limitation, I do not think the paper should be accepted in its current form, but it should be completely revised for future submission. However, as the new results offer more reasonable comparisons, I will leave the final decision to AC on whether such a large modification is acceptable.

**Limitations:**

Yes.

**Paper Formatting Concerns:**

No.

**Quality:**

2

**Strengths And Weaknesses:**

## Strength
- The idea behind Flow-GRPO was clear and well-explained in the paper. The delivery and organization of the paper were concise.
- The experimental results are significantly better than the original pre-trained model, demonstrating the effectiveness of the proposed approach.

## Weakness
- The paper claimed to be the "first method" to incorporate online RL into flow-based models, which I find to be **an overclaim**. For example, the Stable Diffusion 3 paper itself already applied Diffusion DPO [1] to finetune its flow-based text-to-image model. Meanwhile, the online RL methods explicitly designed for flow matching models also exist, e.g., ORW [2] that utilizes the reward-weighted loss. Therefore, the authors should avoid overclaiming. Furthermore, there are abundant approaches in RLHF that are model-agnostic and can be directly applied to align flow matching models in an online setup, e.g., RAFT [3] and DRaFT [4].
- **The contributions seem limited to me**. The proposed method is a simple combination of GPRO and the well-known connection between flow and diffusion models (e.g., [5,6]). The authors also admitted that they directly followed the previous formulation of the conversion between ODE and SDE.
- **RL baselines were very limited**. The authors provided comparisons with existing text-to-image models without such alignment, which are less meaningful in demonstrating the effectiveness of the proposed online RL algorithm. In this way, only the last two rows in Table 1 provided meaningful comparison, which were **insufficient in demonstrating the superior performance over other RL approaches**. Only a **limited comparison with the RL baselines** was provided in the appendix on a single metric. As this paper was not for the benchmark track, such an organization might be misleading for the RL community.

[1] Wallace, Bram, et al. "Diffusion model alignment using direct preference optimization." Proceedings of the IEEE/CVF Conference on Computer Vision and Pattern Recognition. 2024.

[2] Fan, Jiajun, et al. "Online Reward-Weighted Fine-Tuning of Flow Matching with Wasserstein Regularization." arXiv preprint arXiv:2502.06061 (2025).

[3] Dong, Hanze, et al. "Raft: Reward ranked finetuning for generative foundation model alignment." arXiv preprint arXiv:2304.06767 (2023).

[4] Clark, Kevin, et al. "Directly fine-tuning diffusion models on differentiable rewards." arXiv preprint arXiv:2309.17400 (2023).

[5] Albergo, Michael S., Nicholas M. Boffi, and Eric Vanden-Eijnden. "Stochastic interpolants: A unifying framework for flows and diffusions." arXiv preprint arXiv:2303.08797 (2023).

[6] Domingo-Enrich, Carles, et al. "Adjoint matching: Fine-tuning flow and diffusion generative models with memoryless stochastic optimal control." arXiv preprint arXiv:2409.08861 (2024).

---

> ### Author Rebuttal · Authors · 2025-07-31
>
> We sincerely appreciate your insightful comments and efforts in reviewing our manuscript. We respond to each of your comments one-by-one in what follows.
>
> ---
>
> ` [W1] The paper claimed to be the "first method" to incorporate online RL into flow-based models, which I find to be an overclaim. `
>
> We appreciate the reviewer’s comment and agree that it's important to clarify what we mean by “online reinforcement learning” in this context. In our work, we specifically refer to algorithms from the typical reinforcement learning literature, such as PPO [1], GRPO [2], DQN [3], DDPG [4], and SAC [5].
>
> This categorization is consistent with the distinctions made in prior work. For example, the Diffusion-DPO [6] paper explicitly classifies DPOK [7] and DDPO [8] as RL-based methods, while grouping other approaches into separate categories. Furthermore, the original DPO [9] paper characterizes its own contribution as “a simple **RL-free** algorithm for training language models,” reinforcing the distinction between RL-based and non-RL-based alignment methods.
>
> We will revise the manuscript and acknowledge prior work such as Diffusion-DPO [6], ORW [10], RAFT [11], and DRaFT [12], and discuss how they relate to our setting.
>
> [1] Proximal policy optimization algorithms
> [2] DeepSeekMath: Pushing the Limits of Mathematical Reasoning in Open Language Models
> [3] Playing Atari with Deep Reinforcement Learning
> [4] Continuous control with deep reinforcement learning
> [5] Soft Actor-Critic: Off-Policy Maximum Entropy Deep Reinforcement Learning with a Stochastic Actor
> [6] Diffusion Model Alignment Using Direct Preference Optimization
> [7] DPOK: Reinforcement Learning for Fine-tuning Text-to-Image Diffusion Models
> [8] Training Diffusion Models with Reinforcement Learning
> [9] Direct Preference Optimization: Your Language Model is Secretly a Reward Model
> [10] Online Reward-Weighted Fine-Tuning of Flow Matching with Wasserstein Regularization
> [11] Raft: Reward ranked finetuning for generative foundation model alignment
> [12] Directly Fine-Tuning Diffusion Models on Differentiable Rewards
>
>
>
> ---
>
> ` [W2] The contributions seem limited to me. `
>
>
> Thank you for the comment. Our paper does not aim to propose a novel RL algorithm but instead aims to address a critical open question: *Can contemporary generative models be improved through online RL, and how to achieve this for the flow-matching backbones now dominating SOTA generation models.* Before our work, PPO/GRPO have shown remarkable success on understanding tasks with LLMs, while their applicability to flow-matching generation models have yet not been demonstrated. We show that with carefully chosen modifications, flow-matching models can profit from online RL, yielding measurable gains in multi generation tasks. Our results provide key design guidelines and empirical evidence for applying online RL to flow-based generators, opening new directions for future research in this area. We summarise our key contributions as follows:
>
> - **We establish a well-defined MDP for flow-matching generators.** GRPO presupposes stochestic, step-by-step transitions so that a policy $p_{\theta}(a_t \mid s_t)$ is well defined, yet a standard FM model evolves deterministically under its ODE formulation. By leveraging the connections between FM models and score-based generative models, we convert the ODE into an SDE with conytolled noise supplying the required randomness. This injects exactly the stochasticity needed to instantiate the MDP without altering the model's marginal distribution.
> - **We introduce several practical strategies that make online RL both efficient and robust.** We find that online RL does not require the standard long timesteps for training sample collection. By using fewer denoising steps during training and retaining the original steps during testing, we can significantly accelerate the training process with 4x speed up. Additionally, we show that the KL constraint could suppress reward hacking, reaching the high reward while preserving image quality and diversity.
> - **We provide empirical validation of the effectiveness of Flow-GRPO.** We evaluate Flow-GRPO on T2I tasks with various reward types. This includes: 1) verifiable rewards for compositional generation and visual text rendering; 2) model-based rewards for human-preference alignment. Across all settings, Flow-GRPO yields consistent improvements, demonstrating that the proposed framewrok is a promising path for enhancing SOTA generative models.
>
>
>
> ---
>
> ` [W3] Only a limited comparison with the RL baselines was provided in the appendix on a single metric. As this paper was not for the benchmark track, such an organization might be misleading for the RL community `
>
>
>
> Thank you for pointing out this issue! We agree that a broader evaluation is necessary. We have therefore added some extensive experiments that will be included in the revision. These experiments are summarised below:
>
>
> ### 1. Comparison with other RL baselines on GenEval.
>
> Similar to the Figure 6 in the appendix, we compare Flow-GPRO against offline SFT/DPO and their online variants. For each method we track the GenEval evaluation reward and apply early stopping when visual quality collapse is observed. The reward scores, measured at increasing number of training prompts, are list as below:
>
>
> $$
> \begin{array}{lccccccc}
> \hline
> \text{Method} & \text{0} & \text{2000} & \text{5000} & \text{10000} & \text{15000} & \text{20000} & \text{25000} \newline
> \hline
> \textit{Offline Methods} & & & & & & & \newline
> \text{Offline DPO} (\beta = 100) & 0.63 & 0.71 & 0.76 & 0.78 & 0.74 & 0.76 & -  \newline
> \text{Offline DPO} (\beta = 1)   & 0.63 & 0.65 & 0.46 & -    & -    & -    & -  \newline
> \text{Offline SFT}               & 0.63 & 0.72 & 0.74 & 0.74 & 0.76 & 0.75 & 0.75  \newline
> \hline
> \textit{Online Methods} & & & & & & & \newline
> \text{Online DPO} (\beta = 100)  & 0.63 & 0.70 & 0.81 & 0.72 & 0.65 & 0.47 & - \newline
> \text{Online SFT}                & 0.63 & 0.72 & 0.45 & 0.0  & -     & -    & -  \newline
> \textbf{Flow GRPO (ours)}        & \textbf{0.63} & \textbf{0.65} & \textbf{0.72} & \textbf{0.80} & \textbf{0.84} & \textbf{0.92} & \textbf{0.95}  \newline
> \hline
> \end{array}
> $$
>
> We observe that offline baselines like offline DPO and offline SFT hit a ceiling early, with the reward converging at roughly 0.76. Their online variants improve faster but are fragile, which emerges severe reward-hacking and ultimately causes the models to collapse. By contrast, our Flow-GRPO trains smoothly, pushing the reward up to about 0.95 without incuring significant hacking issues.
>
> Note that the reviewer mentioned RAFT: for each input $x_t$, it generates a group of $G$ responses and selects the one with the highest reward $y_t$ within the group for supervised fine-tuning. In fact, this is essentially the same as our baseline labeled "SFT" in the paper.
>
>
>
> ### 2. Comparison with DDPO on pickscore
>
> DDPO was originally designed for diffusion backbones, so we adapted it to flow-matching models by applying our ODE-to-SDE conversion. Using SD-3.5-M as the base model and PickScore as the reward signal, we track the evaluation reward over the entire training process, as shown below:
>
>
> $$
> \begin{array}{l|ccccc}
> \hline
> \text{Method} & \text{Step 0} & \text{Step 240} & \text{Step 480} & \text{Step 720} & \text{Step 960} \newline
> \hline
> \text{DDPO (with ODE-to-SDE)}  & 21.72 & 21.91 & 22.01 & 21.93 & 17.64   \newline
> \textbf{Flow-GRPO} & \textbf{21.72} & \textbf{22.53} & \textbf{23.01} & \textbf{23.31} & \textbf{23.41}   \newline
> \hline
> \end{array}
> $$
>
> We observe that DDPO's reward rises more slowly than Flow-GRPO's and collapses in the late stages, whereas Flow-GRPO trains steadily and continues to improve throughout the training process.
>
> ---
>
> ` [Q1 & Q2] Results for CLIP Score and ORW `
>
> Thank you for the suggestion. Since the official ORW repository only provides a partial implementation, we re-implemented ORW based on their repo within our reward-weighted regression baseline. The loss function is as follows:
>
> ```python
> fm_loss = ((vt - ut) ** 2).mean(dim=(1, 2, 3))
> w2_loss = ((vt - vt_ref) ** 2).mean(dim=(1, 2, 3))
> total_loss = torch.mean(torch.exp(beta * rewards) * fm_loss + alpha * w2_loss)
> ```
>
> **Experimental setup:**
>
> * **Training prompts:** Same as those used in our Human Preference Alignment task.
> * **ORW hyperparameters:** β = 0.5, α = 1 (lower values led to unstable training); `steps_per_epoch` (i.e., how frequently the data-collecting policy is updated) was selected from {20, 40, 100, 400} based on best performance.
>
> **Reward scores** on the test set over training steps:
>
> $$
> \begin{array}{l|ccccc}
> \hline
> \text{Method} & \text{Step 0} & \text{Step 240} & \text{Step 480} & \text{Step 720} & \text{Step 960} \newline
> \hline
> \text{SD3.5-M + ORW}  & 28.79 & 29.05 & 29.15 & 27.58 & 23.05    \newline
> \textbf{SD3.5-M + Flow-GRPO} & 28.79 & \textbf{29.10} & \textbf{29.17} & \textbf{29.51} & \textbf{29.89}   \newline
> \hline
> \end{array}
> $$
>
> Following ORW’s Table 1, we randomly sampled 50 DrawBench prompts and generated 64 images per prompt to compute CLIP and Diversity scores. We used ORW’s best-performing checkpoint on the test set (at step 480). Flow-GRPO outperforms ORW on both metrics:
>
> $$
> \begin{array}{l|cc}
> \hline
> \text{Method} & \text{CLIP Score}\uparrow & \text{Diversity Score}\uparrow \newline
> \hline
> \text{SD3.5-M}  & 27.99 & 0.96   \newline
> \textbf{SD3.5-M + ORW} & 28.40 & 0.97   \newline
> \textbf{SD3.5-M + Flow-GRPO} & \textbf{30.18} & \textbf{1.02}   \newline
> \hline
> \end{array}
> $$
>
> We will include these results in the revision.

---

> > ### Comment · Reviewer_JifQ · 2025-08-04
> >
> > I appreciate the authors' rebuttals with the additional experimental results. While the new results on the comparison with existing online RL approaches make much more sense for a paper targeted at RL, I noted that the current delivery of the paper requires a **significant rewrite** of the whole manuscript, with the new results presented in the main text and current results in Table 1 either discarded or moved to the appendix. I will further elaborate on my points as follows.
> >
> > - Regarding the **overclaim**. First, I did not question the scope of the online RL in this work. Instead, I question the authors' claim to be the *first* in applying online RL approaches to flow-based models. Indeed, **the authors also acknowledged in their rebuttal that there exists previous work**. Therefore, the original paper is overclaiming about this point.
> > - Regarding the **contributions**.
> >   > *We establish a well-defined MDP for flow-matching generators.*
> >
> >   Stochasticity is not mandatory for defining an MDP. The existing work (e.g., RAFT, DRaFT, and ORW) also defined (though maybe implicitly) a one-step MDP such that the reward weighting techniques can be applied.
> >   > *We introduce several practical strategies that make online RL both efficient and robust.*
> >
> >   I did not see any comparison in the original manuscript with other online RL approaches. Without even a comparison, it is unclear to which baseline model the proposed approach is "efficient and robust".
> >   > *We provide empirical validation of the effectiveness of Flow-GRPO.*
> >
> >   Same as above. It is unclear whether the proposed approach is effective compared to existing online RL methods, as none were compared in the original manuscript.
> >
> >   In conclusion, the **theoretical contributions are minor**, which was also acknowledged by the authors in their rebuttal. This could have been neglected if the authors had provided more convincing and comprehensive practical evaluations of existing online RL approaches. Sadly, and surprisingly, as a paper that champions its RL contributions in the title, this part is **completely missing from the original manuscript**. Such an issue was also identified by Reviewer L1rz.
> >
> > Therefore, based on the above justifications, although I do not question the performance of the proposed method given the new results, I will maintain my original overall score and suggest a complete reorganization of the paper for a future submission. I will modify the other scores accordingly.

---

> > > ### Author Response · Authors · 2025-08-05
> > > **Response to Reviewer JifQ**
> > >
> > > Thank you again for your thoughtful comments. We are glad our rebuttal has addressed your concerns about the performance. For the remaining issues, we respond point-by-point below.
> > >
> > > ---
> > >
> > > > [W1] Regarding the overclaim. First, I did not question the scope of the online RL in this work. Instead, I question the authors' claim to be the first in applying online RL approaches to flow-based models. Indeed, the authors also acknowledged in their rebuttal that there exists previous work. Therefore, the original paper is overclaiming about this point.
> > >
> > > Although DPO-like and RWR-like methods are often associated with reinforcement learning, they are not strictly reinforcement learning algorithms. For example, DPO[1] explicitly claims in its original paper that it is "RL-free". To ensure precision, we will describe our work as the first to apply **online policy gradient approaches to flow-based models** in revision.
> > >
> > > [1]. Direct Preference Optimization: Your Language Model is Secretly a Reward Model, NeurIPS 2023.
> > >
> > > ---
> > >
> > > > [W2-1] Stochasticity is not mandatory for defining an MDP. The existing work (e.g., RAFT, DRaFT, and ORW) also defined (though maybe implicitly) a one-step MDP such that the reward weighting techniques can be applied.
> > >
> > > We fully acknowledge that stochasticity is not necessary to define an MDP in the single-step case. That formulation can still be used to optimize methods such as ORW[2]. Our intention was to highlight that by converting the ODE into an SDE, we introduce stochasticity that **makes policy gradient training applicable**.
> > >
> > > [2]. Online Reward-Weighted Fine-Tuning of Flow Matching with Wasserstein Regularization, ICLR 2025.
> > >
> > > ---
> > >
> > > > [W2-2 & W2-3] I did not see any comparison in the original manuscript with other online RL approaches. Without even a comparison, it is unclear to which baseline model the proposed approach is "efficient and robust".
> > >
> > > Thank you for the comment. We would like to clarify that **the original manuscript does include comprehensive comparisons with baselines**, including SFT, RWR, DPO, as well as their corresponding online variants. In particular, we have now added explicit experimental comparisons with ORW, as requested. **These are straightforward extensions of the existing experiments rather than a significant rewrite.**
> > >
> > > Furthermore, our claim of efficiency and robustness is supported by two key contributions described in the manuscript: (1) denoising step reduction, which significantly accelerates training for Flow-GRPO as show in Figure 5, and (2) the use of a KL penalty to mitigate reward hacking as shown in Figure 4.

---

> > > > ### Comment · Reviewer_JifQ · 2025-08-06
> > > >
> > > > I thank the authors again for their detailed clarifications. I would first like to point out that I did acknowledge the new results in my previous comment, as they provide far more meaningful and fair comparisons to the RL baselines. My concern is that, according to my understanding of the NeurIPS guideline, the revision during the rebuttal period should not be significantly different from the original submission. Based on our previous discussion, I believe we all agree that the original manuscript is problematic in its delivery and claimed contributions, therefore requiring a significant rewrite and reorganization. Nonetheless, I will leave the final decision to the AC on whether such a large modification of the manuscript is acceptable.

---

### Official Review · Reviewer_cqnV · 2025-07-03

**Clarity:** 3
**Significance:** 3
**Originality:** 2
**Rating:** 4
**Confidence:** 4

**Summary:**

This paper presents Flow-GRPO, a method that enables online reinforcement learning for flow matching generative models by introducing an ODE-to-SDE conversion and a denoising reduction strategy. The approach achieves strong improvements in compositional image generation, text rendering, and human preference alignment on several benchmarks, while preserving image quality and diversity.

**Questions:**

See weakness

**Ethical Concerns:**

["NO or VERY MINOR ethics concerns only"]

**Final Justification:**

My questions have been resolved. However, I still find the novelty limited, as there are no new theoretical contributions in applying GRPO to flow matching models.

**Limitations:**

See weakness

**Quality:**

3

**Strengths And Weaknesses:**

**Strength:**

1. This paper proposes to apply online RL to flow matching models by the ODE-to-SDE conversion to introduce essential stochasticity for RL.

2. Denoising reduction greatly speeds up training without sacrificing output quality and show substantial gains across standard T2I benchmarks, with thorough ablations on reward hacking and generalization.


**Weakness:**

1. The conversion from Flow ODE to SDE is not a novel contribution, as stochastic sampling for flow-based models has already been explored in prior works such as [1].

2. The paper’s main contributions are: (a) transferring Flow ODE to SDE, which is well-studied in prior literature, and (b) applying GRPO to flow models. Since GRPO has already been extensively explored in the context of LLMs, this work appears to be largely an application of existing techniques to diffusion models, which limits its novelty. How does the exploration-exploitation tradeoff in the Flow-GRPO compare with diffusion models or standard RL settings?

[1] SiT: Exploring Flow and Diffusion-based Generative Models with Scalable Interpolant Transformers

Other questions:

1. The alation on the group size $G$ on GRPO.

---

> ### Author Rebuttal · Authors · 2025-07-31
>
> We sincerely appreciate your insightful comments and efforts in reviewing our manuscript. We respond to each of your comments one-by-one in what follows.
>
>
> ---
>
> `  [W1] The conversion from Flow ODE to SDE is not a novel contribution, as stochastic sampling for flow-based models has already been explored in prior works. `
>
>
>
> Thank you for raising this concern. Our ODE-to-SDE conversion is not intended as a standalone theoretical contribution, but as a practical tool that equips Flow-GRPO with the stochastic transitions required for a valid MDP. Applying GRPO-style optimization in FM models demands a sampling procedure whose step-by-step randomness does not disturb the model's marginal distribution. We therefore construct an SDE that is distribution-equivalent to the original ODE, which is largely inspired by the prior flow-matching theory [1] and shares a similar formulation with prior works such as [2]. **The contribution lies not in the mathematics of the conversion itself, but in adapting and repurposing it to make online RL feasible and empirically effective for contemporary flow-matching backbones.**
>
>
> [1]. Adjoint Matching: Fine-tuning Flow and Diffusion Generative Models with Memoryless Stochastic Optimal Control. ICLR 2025.
> [2]. SiT: Exploring Flow and Diffusion-based Generative Models with Scalable Interpolant Transformers. ECCV 2024.
>
> ---
>
> `  [W2-1] This work appears to be largely an application of existing techniques to diffusion models, which limits its novelty. `
>
>
> Thank you for the comment. Our paper does not aim to propose a novel RL algorithm but instead aims to address a critical open question: *Can contemporary generative models be improved through online RL, and how to achieve this for the flow-matching backbones now dominating SOTA generation models.* Before our work, PPO/GRPO have shown remarkable success in understanding tasks with LLMs, while their applicability to flow-matching generation models has not yet been demonstrated. We show that with carefully chosen modifications, flow-matching models can profit from online RL, yielding measurable gains in multi generation tasks. Our results provide key design guidelines and empirical evidence for applying online RL to flow-based generators, opening new directions for future research in this area. We summarise our key contributions as follows:
>
> - **We establish a well-defined MDP for flow-matching generators.** GRPO presupposes stochastic, step-by-step transitions so that a policy $p_{\theta}(a_t \mid s_t)$ is well defined, yet a standard FM model evolves deterministically under its ODE formulation. By leveraging the connections between FM models and score-based generative models, we convert the ODE into an SDE with controlled noise supplying the required randomness. This injects exactly the stochasticity needed to instantiate the MDP without altering the model's marginal distribution.
> - **We introduce several practical strategies that make online RL both efficient and robust.** We find that online RL does not require the standard long timesteps for training sample collection. By using fewer denoising steps during training and retaining the original steps during testing, we can significantly accelerate the training process with 4x speedup. Additionally, we show that the KL constraint could suppress reward hacking, reaching the high reward while preserving image quality and diversity.
> - **We provide empirical validation of the effectiveness of Flow-GRPO.** We evaluate Flow-GRPO on T2I tasks with various reward types. This includes: 1) verifiable rewards for compositional generation and visual text rendering; 2) model-based rewards for human-preference alignment. Across all settings, Flow-GRPO yields consistent improvements, demonstrating that the proposed framework is a promising path for enhancing SOTA generative models.
>
>
> ---
>
> `  [W2-2] How does the exploration-exploitation tradeoff in the Flow-GRPO compare with diffusion models or standard RL settings? `
>
> Thank you for the question. In Flow-GRPO, the exploration–exploitation tradeoff is governed by the noise scale $\sigma_t$ in Equation (9). Conceptually, $\sigma_t$ plays a role similar to the $\epsilon$ in $\epsilon$-greedy strategies in standard RL settings: a larger $\sigma_t$ increases exploration by introducing greater stochasticity into the denoising process, but at the cost of reduced policy quality and potential instability or suboptimal behaviors.
>
> We provide an empirical analysis of this tradeoff in the paragraph starting at line 225, where we systematically vary $\sigma_t$ and study its effect on final performance. The results demonstrate a clear tradeoff, highlighting the importance of tuning $\sigma_t$ appropriately to balance exploration and exploitation.
>
>
> ---
>
> ` [W3] The ablation on the group size G on GRPO. `
>
>
> Thank you for this valuable suggestion. We have added an ablation study on the group size $G$ using PickScore as the reward function. When we reduced the group size to $G=12$ and $G=6$, training became unstable and collapsed at step 778 and step 742, respectively. In contrast, $G=24$ remained stable throughout the full process. We logged the PickScore on the test set before and after the collapse, as shown below.
>
>
> $$
> \begin{array}{l|ccc}
> \hline
> \textbf{Group Size} & \textbf{Step 0} & \textbf{Step 720} & \textbf{Step 956} \newline
> \hline
> G=24 & 21.72 & 23.31 & 23.41  \newline
> G=12 & 21.72 & 22.95 & 20.96  \newline
> G=6  & 21.72 & 22.64 & 20.68  \newline
> \hline
> \end{array}
> $$
>
> Our conclusion is that a small group size leads to inaccurate advantage estimates, inflating the variance and eventually triggering collapse. A similar phenomenon is also reported in [3, 4]. We will add these results in the revision.
>
> [3] ProRL: Prolonged Reinforcement Learning Expands Reasoning Boundaries in Large Language Models. Arxiv 2025.
> [4] AceReason-Nemotron: Advancing Math and Code Reasoning through Reinforcement Learning. Arxiv 2025.

---

> > ### Comment · Reviewer_cqnV · 2025-08-05
> >
> > Thanks for your reply. My concerns have been solved.

---

> > > ### Author Response · Authors · 2025-08-06
> > >
> > > Thank you so much for your quick response and for helping improve our work. We will incorporate your suggestions, including an explanation of the exploration-exploitation tradeoff and an ablation study on group size, into the final version.

---

### Official Review · Reviewer_UUcc · 2025-07-03

**Clarity:** 2
**Significance:** 3
**Originality:** 2
**Rating:** 5
**Confidence:** 2

**Summary:**

This paper proposes a method to fine-tune a pre-trained Flow-Matching (FM) sampler via online reinforcement learning (RL). By leveraging the SDE formulation of FM, the authors successfully reduce the number of FM sampling steps required at inference time.

**Questions:**

- **Q1**: Why choose flow matching as the base model? How would the approach differ if a diffusion model were used?
- **Q2**: As I am not specialized in this area, could you explain the design principles behind the reward function $f$ defined on line 119?
- **Q3**: Again, from a non-expert perspective, what are the main factors limiting the scalability of the proposed method?

**Ethical Concerns:**

["NO or VERY MINOR ethics concerns only"]

**Final Justification:**

## Resolved Issues

- **Q1 (Choice of Flow Matching):**
  Authors explained that flow-matching is now the leading backbone in SOTA image/video generators, motivating direct post-training with RL.

- **Q2 (Reward Function Design):**
  Clarified that `f` is the GRPO surrogate objective, and its clipping-based trust region, advantage weighting, and KL penalty jointly ensure stable, monotonic policy improvement.

- **Q3 (Scalability Factors):**
  Denoising-reduction for ~4× speed-up and the KL term’s role in curbing reward hacking are now clearly articulated.

## Remaining Concerns

- The core idea—injecting stochasticity via an SDE formulation and applying a standard GRPO update—is relatively simple. Some readers may seek deeper theoretical novelty.

## Weighting of Aspects

- **Empirical Validation (High weight):**
  Comprehensive experiments (GenEval, text rendering, human preferences) convincingly demonstrate effectiveness.

- **Clarity (Medium weight):**
  Rebuttal improved my understanding; clarity score raised to 3.

- **Originality (Low–Medium weight):**
  While straightforward, the work fills an important gap by validating RL on flow-matching samplers; originality score raised to 3.

**Limitations:**

yes

**Quality:**

3

**Strengths And Weaknesses:**

**Strengths**
- **Quality**: The utility of the method is supported by comprehensive, multi-faceted experiments.
- **Clarity**: The core ideas are presented in a clear and understandable manner.
- **Significance**: Fine-tuning generative samplers offline via RL is novel, and the fact that training steps can be drastically reduced without loss of performance is particularly interesting.

**Weaknesses**
- **Originality**: For readers already familiar with FM theory, the idea may appear straightforward.

---

> ### Author Rebuttal · Authors · 2025-07-31
>
> We sincerely appreciate your insightful comments and efforts in reviewing our manuscript. We respond to each of your comments one-by-one in what follows.
>
>
> ---
>
>
> ` [W1] About the originality; `
>
>
> Thank you for the comment. Our paper does not aim to propose a novel RL algorithm but instead aims to address a critical open question: *Can contemporary generative models be improved through online RL, and how to achieve this for the flow-matching backbones now dominating SOTA generation models.* Before our work, PPO/GRPO have shown remarkable success in understanding tasks with LLMs, while their applicability to flow-matching generation models has not yet been demonstrated. We show that with carefully chosen modifications, flow-matching models can profit from online RL, yielding measurable gains in multi generation tasks. Our results provide key design guidelines and empirical evidence for applying online RL to flow-based generators, opening new directions for future research in this area. We summarise our key contributions as follows:
>
> - **We establish a well-defined MDP for flow-matching generators.** GRPO presupposes stochastic, step-by-step transitions so that a policy $p_{\theta}(a_t \mid s_t)$ is well defined, yet a standard FM model evolves deterministically under its ODE formulation. By leveraging the connections between FM models and score-based generative models, we convert the ODE into an SDE with controlled noise supplying the required randomness. This injects exactly the stochasticity needed to instantiate the MDP without altering the model's marginal distribution.
> - **We introduce several practical strategies that make online RL both efficient and robust.** We find that online RL does not require the standard long timesteps for training sample collection. By using fewer denoising steps during training and retaining the original steps during testing, we can significantly accelerate the training process with 4x speedup. Additionally, we show that the KL constraint could suppress reward hacking, reaching the high reward while preserving image quality and diversity.
> - **We provide empirical validation of the effectiveness of Flow-GRPO.** We evaluate Flow-GRPO on T2I tasks with various reward types. This includes: 1) verifiable rewards for compositional generation and visual text rendering; 2) model-based rewards for human-preference alignment. Across all settings, Flow-GRPO yields consistent improvements, demonstrating that the proposed framework is a promising path for enhancing SOTA generative models.
>
>
> ---
>
> ` [Q1-1] Why choose flow matching as the base model?`
>
> We choose flow matching models because it provides faster sampling speed at comparable quality. Contemporary leading image generation models (e.g., SD3 [1], FLUX [2]) and video generation models (e.g., Sora [3], Wan2.1 [4], Seedance [5]) now adopt flow matching in place of traditional diffusion, which means our method could apply directly to their post‑training stage.
>
> [1]. Scaling Rectified Flow Transformers for High-Resolution Image Synthesis.
> [2]. FLUX.
> [3]. Sora: Video generation models as world simulators.
> [4]. Wan: Open and Advanced Large-Scale Video Generative Models.
> [5]. Seedance 1.0: Exploring the Boundaries of Video Generation Models.
>
> ---
>
> ` [Q1-2] How would the approach differ if a diffusion model were used`
>
> Thanks for asking this clarification question!
>
> 1. **GRPO requires an MDP with stochastic step-to-step transitions so that a policy $p_{\theta}(a_t \mid s_t)$ is well defined.** A flow-matching model, in its standard ODE form, is purely deterministic, and there is no randomness in a single denoising step. To address it, we therefore convert the ODE to an equivalent SDE to inject randomness, creating the stochastic transitions needed to instantiate the MDP.
> 2. **Our proposed method could be applied directly in diffusion models without modification.** If a diffusion model were used instead, this stochasticity is already built in their DDPM-style sampling, so the same pipeline in Flow-GRPO can be directly utilized in diffusion models' optimization.
>
>
>
> ---
>
> ` [Q2] The design principles behind the reward function f defined on line 119? `
>
>
> Thank you for the question. There seems to be a misunderstanding: in our notation, `f` refers to the GRPO objective function, not the reward model (which is denoted by `r`).  A proper reward model typically needs to assign a scalar score to each generated sample (higher means better), and does not need to be differentiable. The three reward functions we adopt in the manuscript are defined on Line 168, Line 169, and Line 176.
>
> The GRPO objective (`f`) is designed to ensure stable and monotonic policy improvement through three key principles:
>
> 1. **Trust Region.** The core of the function is a clipping mechanism that uses `min` and `clip` functions to limit the magnitude of policy updates. This creates a **trust region**, allowing only small, safe adjustments around the current policy. This effectively prevents performance collapse due to overly large update steps and ensures training **stability**.
>
> 2. **Advantage-Weighted Updates.** Our goal is to maximize the objective `f`, which corresponds to maximizing $r_t Â_t$. Therefore, the direction of updates is guided by the **advantage function** $Â_t$:
>
>  * When $Â_t > 0$ (i.e., the action is better than average), the objective reduces to maximizing $r_t$. Since the denominator of $r_t$ has no gradient, this effectively means increasing the numerator — that is, increasing the probability of that action.
>  * When $Â_t < 0$ (i.e., the action is worse than average), the probability of that action is decreased.
>
> 3. **KL Divergence Regularization.** The loss function includes a KL divergence penalty term, $-β D_{KL}(π_θ || π_{ref})$. It acts as a **soft constraint**, preventing the new policy $π_θ$ from deviating too far from a known reference policy $π_{ref}$ (often the pretrained policy).
>
>
>
> ---
>
>
> ` [Q3] What are the main factors limiting the scalability of the proposed method? `
>
> Thank you for raising this point. We discuss two main factors that govern scalability.
>
> 1. **Efficiency.** Scaling the post-training stage requires a more efficient training strategy. Our manuscript introduces a denoising reduction strategy that trains the model with only a quarter of the normal sampling steps, achieving roughly a 4x speed-up with very subtle loss in quality. Further works could attain gains by adopting a dynamic data selection, for instance, periodically discarding samples whose advantage estimates show low variance.
> 2. **Avoid Reward Hacking.** Reward hacking would become increasingly problematic as the training scale grows. We observe that a KL regulariser can curb the hacking effectively, allowing more optimisation steps before the model starts to hack the reward function (see Figure 8). Even so, an ultimate safeguard is a robust and accurate reward model, which makes it harder for the generator to discover loopholes. Further work on powerful, ensemble-based rewards will therefore be one key to scaling the algorithm.

---

> > ### Comment · Reviewer_UUcc · 2025-08-01
> >
> > Thank you for your detailed rebuttal. In particular:
> >
> > Q2 (Reward Function Design): Your explanation clarified that f is the GRPO surrogate objective (not the reward model) and that its clipping-based trust region, advantage-weighted updates, and KL regularization together ensure stable, monotonic policy improvement.
> >
> > Q3 (Scalability Factors): Your discussion of denoising-reduction for a 4× training speed-up and the role of the KL penalty (and ultimately robust reward models) in curbing reward hacking makes the method’s scaling limits much clearer.
> >
> > I now appreciate that, although the core idea—injecting stochasticity into a deterministic flow-matching sampler via an SDE formulation and then applying a standard GRPO update—is relatively simple, the thorough experimental validation convincingly demonstrates its effectiveness. Accordingly, I am raising my scores.

---

> > > ### Author Response · Authors · 2025-08-01
> > > **Response to Reviewer UUcc**
> > >
> > > Thank you so much for your quick response and for helping improve our work. We will make sure to incorporate your suggestions, including a physical interpretation of the GRPO surrogate objective and a discussion of the main factors limiting the scalability of Flow GPPO, into the final version.

---

### Official Review · Reviewer_2Psq · 2025-07-05

**Clarity:** 4
**Significance:** 4
**Originality:** 4
**Rating:** 6
**Confidence:** 4

**Summary:**

- This work proposes Flow-GRPO, the first method to perform online RL in flow matching models.
- There are two problems to applying online RL methods like GRPO directly to flow matching models. 1) Flow matching models rely on a deterministic process based on ODE. As a result, the sampling during inference is not stochastic while online RL methods rely on exploration i.e. stochasticity during inference and 2) Online RL methods rely on efficient sampling to collect training data. Flow models inherently require multiple denoising steps during inference for 1 generation. This affects the efficiency of online RL methods.
- Flow GRPO solves both these problems by 1) converting the **ODE to an SDE** thereby making the inference stochastic and 2) applying **Denoising reduction** strategy. The denoising reduction strategy uses fewer de-noising steps during training but uses the full schedule during inference.
- Authors show that unlike other online RL methods, Flow-GRPO shows minimal reward hacking and gets almost perfect score on GenEval outperforming GPT-4o. On visual text-rendering, SD3-M's accuracy increases from 59% to 92%.

**Questions:**

- Effect of ODE-to-SDE conversion. How much does this affect metrics like FID/SSIM/ human evaluations?
- Why does using fewer steps improve reward? In Fig. 5a, Step=10 outperforms Step=40 in GenEval score as well. I understand that speed is definitely improved with 4x reduction in steps but the reward also doesn't go above 0.9 with Steps=40? Is it because of the number of updates? Is the graph adjusted for the number of updates under each setting?
- How strong is the generalization? In Table-4, authors show that the model generalizes well for objects > 4. When does it break?
- All the Text rendering examples show text with few words. Will the model be robust to long visual text as well? When does the model break?

**Ethical Concerns:**

["NO or VERY MINOR ethics concerns only"]

**Final Justification:**

I thank the authors for addressing my questions. Authors have satisfactorily addressed my questions. After going through other reviews and author responses, I vote to retain my initial rating.

**Limitations:**

yes.

**Paper Formatting Concerns:**

No.

**Quality:**

4

**Strengths And Weaknesses:**

**Strengths**
- This paper proposes a solution to a very hard and important problem, applying online RL to diffusion/flow matching methods and shows that, similar to text generation, online RL improves media generation significantly as well. This work would be very useful to the community.
- The paper is well written and has enough details for proper reproduction.
- Qualitative and quantitative results are impressive.

**Weaknesses**
- I see tremendous value in the current work for media generation community and I don't see any potential weaknesses of this work.

---

> ### Author Rebuttal · Authors · 2025-07-31
>
> We sincerely appreciate your insightful comments and efforts in reviewing our manuscript. We respond to each of your comments one-by-one in what follows.
>
> ---
>
> ` [Q1] How does this ODE-to-SDE conversion affect metrics like FID/SSIM/ human evaluations? `
>
>
> We evaluate the impact of converting ODE sampling to SDE sampling using SD‑3.5‑M as the base model. We compared 40-step ODE and 40-step SDE sampling on MSCOCO‑2014 for FID/IS/CLIP Score, and on DrawBench for aesthetic/DEQA/ImageReward/PickScore/UnifiedReward. The two settings yield essentially the same performance at 40 steps, which aligns with our derivation in Sec. A showing that both share the same marginal distribution. For clarity, SDE sampling is only used during training; all reported metrics are computed from 40-step ODE samples. Consequently, the ODE-to-SDE conversion has no material impact on FID/ID or human-preference metrics in our experiments.
>
>
>
> $$
> \begin{array}{l|ccc|ccccc}
> \hline
>  & & \textbf{MSCOCO-2014} & & & & \textbf{DrawBench} & & \newline
> \textbf{Model} & \textbf{FID ↓} & \textbf{IS ↓} & \textbf{CLIP Score ↑} & \textbf{Aesthetic ↑} & \textbf{Deqa ↑} & \textbf{ImageReward ↑} & \textbf{PickScore ↑} & \textbf{UnifiedReward ↑} \newline
> \hline
> \text{ODE, 40 steps} & 17.50 & 40.81 & 31.95 & 5.39 & 4.07 & 0.87 & 22.34 & 3.33 \newline
> \text{SDE, 40 steps} & 16.43 & 41.22 & 31.70 & 5.36 & 4.01 & 0.85 & 22.34 & 3.33 \newline
> \hline
> \end{array}
> $$
>
>
>
> ---
>
> ` [Q2-1] Why does using fewer steps improve reward? `
>
>
> Thank you for the question. We view denoising reduction as a strategy to boost efficiency, not to raise the ultimate reward ceiling. In addition to Figure 5, Section C.2 in the Appendix extended ablations on visual text rendering and human-preference alignment. Across these three tasks, a 10-step schedule converges to similar final rewards compared to the 40-step baseline while cutting wall-clock training time by roughly 4x. The higher reward observed therefore reflects faster convergence, not a higher performance.
>
> ---
>
> ` [Q2-2] The number of updates in the graph `
>
> In Figure 5, we measure progress by GPU‑hours, i.e., wall‑clock cost. If we re‑index the curves by the number of optimizer updates, the 10‑step and 40‑step schedules are nearly equivalent. One difference is that each update in the 40-step schedule requires four times the accumulation of the 10‑step schedule's, owing to its 4x longer denoising steps.
>
>
>
>
> ---
>
> ` [Q3] How strong is the generalization? In Table 4, the authors show that the model generalizes well for objects > 4. When does it break? `
>
> Thank you for the suggestion. Following the protocol of Table 4, we extend the generalization test and require the model to generate images with 8/10/12 objects, an extremely challenging setting in which the base SD-3.5-M model almost never succeeds. Flow-GRPO still achieves a substantial improvement, although the absolute success rate naturally decreases as the task becomes more difficult. The results are summarised below:
>
>
> $$
> \begin{array}{l|ccc}
> \hline
> \text{Method} & \text{8 objects} & \text{10 objects} & \text{12 objects} \newline
> \hline
> \text{SD3.5-M}  & 0.02 & 0.01 & 0.02  \newline
> \textbf{SD3.5-M + Flow-GRPO}  & \textbf{0.19} & \textbf{0.07} & \textbf{0.12}  \newline
> \hline
> \end{array}
> $$
>
> ---
>
> ` [Q4] Will the model be robust to long visual text as well? When does the model break? `
>
>
> Thank you for the suggestion. Our text-rendering checkpoint was trained on short phrases (2-4 words), so we further evaluate its generalization on prompts containing a full 10-word sentence. For illustration, a representative long‑text prompt is:
>
> > The final, unplaced piece of a giant, intricate jigsaw puzzle, held in a hand above the puzzle. The piece itself has the words, “The final piece that makes the entire picture make perfect sense.
>
>
>
> In this harder setting the base SD-3.5-M model degrades sharply in accuracy, whereas Flow-GRPO maintains most of its performance. The results are summarised below:
>
>
> $$
> \begin{array}{l|cc}
> \hline
> \text{Method} & \text{2-4 words} & \text{10 words} \newline
> \hline
> \text{SD3.5-M}  & 0.59 & 0.28   \newline
> \textbf{SD3.5-M + Flow-GRPO} & \textbf{0.92} & \textbf{0.82}   \newline
> \hline
> \end{array}
> $$

---

> ### Author Response · Authors · 2025-08-08
>
> We sincerely thank Reviewer 2Psq for your thoughtful and detailed review. We are very pleased that you recognize that our work shows online RL can improve media generation significantly, and that **you found our paper has enough details for proper reproduction and is very useful to the community**.
>
> Your comments regarding FID/SSIM, performance with fewer steps, and generalization have been instrumental in helping us refine our work. To that end, we have provided detailed responses in our rebuttal, including additional experiments specifically designed to address your concerns.
>
> With the discussion phase drawing to a close, we would be grateful if you could let us know whether our clarifications and new results have resolved your initial concerns. We greatly appreciate your time and would welcome any further feedback.

---

### Note · Authors · 2025-08-12

Dear Reviewers and Area Chairs,

We sincerely thank all reviewers for their thoughtful feedback and constructive discussion. We are encouraged that the reviewers have either raised their scores or maintained their current ratings, especially by Reviewer L1rz’s increase from 2 to 5. Reviewer 2Psq recognized the tremendous value of this work for the media generation community; Reviewer UUcc found our method novel, with core ideas that are simple yet effective; Reviewer cqnV noted that our proposed denoising reduction greatly accelerates training without sacrificing output quality; Reviewer JifQ found the idea clear and well-explained in the paper; and Reviewer L1rz affirmed that our method is theoretically sound and demonstrates strong generalization. We greatly appreciate the constructive spirit demonstrated throughout the discussion.

In our rebuttal, we have carefully addressed every reviewer's question in detail, focusing on: (1) correcting misunderstandings; (2) clarifying our technical contributions; and (3) expanding the performance comparisons. To further clarify our work, we have also conducted and included a substantial set of additional experiments and analyses. We are grateful for all feedback, as it has improved the paper’s clarity and strengthened our arguments.

After the discussion, we are glad that all technical concerns have been resolved. The only remaining point is the paper's *"problematic delivery"* raised by Reviewer JifQ. While the additional results have addressed the reviewer's performance concern, the reviewer suggested a significant rewrite to incorporate them. We are grateful for the feedback but respectfully disagree with this assessment, as these new results are natural extensions of comparisons already included in the appendix. With the extra page allowed for the camera-ready version, we will move these experiments into the main paper, which will directly resolve the delivery concern. The revision will strengthen the presentation and clarity without changing the core technical contribution.

We sincerely hope that our responses and new evidence convey both the insight and the potential impact of our paper, and we are grateful for the reviewers' valuable feedback that guided these improvements. Thank you again for your time and thoughtful consideration.

Best regards,

Authors of paper #3110

---

### Decision · Program_Chairs · 2025-09-17

**Decision:**

Accept (poster)

**Comment:**

The authors introduce Flow-GRPO, an application of GRPO to perform preference alignment on flows, which is claimed to be the first application of an online RL algorithm to flows. As policy gradient algorithms require a stochastic policy, an ODE-to-SDE conversion is used define a stochastic sampler for the flow to use with GRPO. Denoising Reduction strategy which involves using fewer denoising steps when running GRPO and using more at evaluation time is introduced to increase training efficiency. Empirically, Flow-GRPO is shown to significantly improve text-to-image generation across multiple tasks, including compositional generation, visual text rendering, and human preference alignment. The method also demonstrates minimal reward hacking due to KL regularization and strong generalization capabilities.

Strengths:
- A simple and general way of applying online RL to flow matching models.
- The quantitative and qualitative results across diverse text-to-image tasks, outperforming GPT-4o, highlight its effectiveness.
- Flow-GRPO is shown to have some generalization capabilities to unseen object categories and counts, and robustness to reward hacking.

Weaknesses:
- Limited novelty, as both the ODE-to-SDE conversion and GRPO are existing techniques.
- Reviewers have taken issue with the claim that Flow-GRPO was the first online RL method applied to flows and the lack of insufficient discussion of prior work on preference alignment methods for flows, though the authors' response has mostly addressed this.
- The lack comparison to any preference alignment baselines in the body of the paper weakens it considerably by leaving some key questions unanswered. There were some such results in the supplementary material and more were provided in response to the reviewers, but they really need to be integrated into the paper.
- All the results in the paper are based on single runs, which is not a good practice, especially for RL algorithms.

The paper in its current form does not meet the bar for acceptance due to missing key baselines and lack of clarity about prior work, but with these issues addressed as promised in the discussion with the reviewers, it would make for a solid contribution to the literature.